# Coding with transient trajectories in recurrent neural networks

**Giulio Bondanelli** *, **Srdjan Ostojic**

Laboratoire de Neurosciences Cognitives et Computationelles, Département d'Études Cognitives, École Normale Supérieure, INSERM U960, PSL University, Paris, France

* giulio.bondanelli@ens.fr

**Data Availability Statement:** All relevant data are within the manuscript and its Supporting Information files.

**Funding:** This work was funded by the Programme Emergences of City of Paris, Agence Nationale de la Rechere grant ANR-16-CE37-0016, and the

## Abstract

Following a stimulus, the neural response typically strongly varies in time and across neurons before settling to a steady-state. While classical population coding theory disregards the temporal dimension, recent works have argued that trajectories of transient activity can be particularly informative about stimulus identity and may form the basis of computations through dynamics. Yet the dynamical mechanisms needed to generate a population code based on transient trajectories have not been fully elucidated. Here we examine transient coding in a broad class of high-dimensional linear networks of recurrently connected units. We start by reviewing a well-known result that leads to a distinction between two classes of networks: networks in which all inputs lead to weak, decaying transients, and networks in which specific inputs elicit amplified transient responses and are mapped onto output states during the dynamics. Theses two classes are simply distinguished based on the spectrum of the symmetric part of the connectivity matrix. For the second class of networks, which is a sub-class of non-normal networks, we provide a procedure to identify transiently amplified inputs and the corresponding readouts. We first apply these results to standard randomly-connected and two-population networks. We then build minimal, low-rank networks that robustly implement trajectories mapping a specific input onto a specific orthogonal output state. Finally, we demonstrate that the capacity of the obtained networks increases proportionally with their size.

## Author summary

Classical theories of sensory coding consider the neural activity following a stimulus as constant in time. Recent works have however suggested that the temporal variations following the appearance and disappearance of a stimulus are strongly informative. Yet their dynamical origin remains little understood. Here we show that strong temporal variations in response to a stimulus can be generated by collective interactions within a network of neurons if the connectivity between neurons satisfies a simple mathematical criterion. We moreover determine the relationship between connectivity and the stimuli that are represented in the most informative manner by the variations of activity, and estimate the

program "Investissements d'Avenir" launched by the French Government and implemented by the ANR, with the references ANR-10- LABX-0087 IEC and ANR-11-IDEX-0001-02 PSL Research University. The funders had no role in study design, data collection and analysis, decision to publish, or preparation of the manuscript.

**Competing interests:** The authors have declared that no competing interests exist.

number of different stimuli a given network can encode using temporal variations of neural activity.

## Introduction

The brain represents sensory stimuli in terms of the collective activity of thousands of neurons. Classical population coding theory describes the relation between stimuli and neural firing in terms of tuning curves, which assign a single number to each neuron in response to a stimulus [1–3]. The activity of a neuron following a stimulus presentation typically strongly varies in time and explores a range of values, but classical population coding typically leaves out such dynamics by considering either time-averaged or steady-state firing.

In contrast to this static picture, a number of recent works have argued that the temporal dynamics of population activity may play a key role in neural coding and computations [4–14]. As the temporal response to a stimulus is different for each neuron, an influential approach has been to represent population dynamics in terms of temporal trajectories in the neural state space, where each axis corresponds to the activity of one neuron [15–18]. Coding in this high-dimensional space is typically examined by combining linear decoding and dimensionality-reduction techniques [19–21], and the underlying network is often conceptualised in terms of a dynamical system [18, 22–30]. Such approaches have revealed that the discrimination between stimuli based on neural activity can be higher during the transient phases than at steady state [16], arguing for a coding scheme in terms of neural trajectories. A full theory of coding with transient trajectories is however currently lacking.

To produce useful transient coding, the trajectories of neural activity need to satisfy at least three requirements [4]. They need to be (i) stimulus-specific, (ii) robust to noise and (iii) non-monotonic, in the sense that the responses to different stimuli differ more during the transient dynamics than at steady-state. This third condition is crucial as otherwise coding with transients can be reduced to classical, steady-state population coding. Recent works have shown that recurrent networks with so-called non-normal connectivity can lead to amplified transients [28, 31–35], but general sufficient conditions for such amplification were not given. We start by reviewing a well-known result linking the norm of the transient activity to the spectrum of the symmetric part of the connectivity matrix. This results leads to a simple distinction between two classes of networks: networks in which all inputs lead to weak, decaying transients, and networks in which specific inputs elicit transiently amplified responses. We then characterize inputs that lead to non-monotonic trajectories, and show that they induce transient dynamics that map inputs onto orthogonal output directions. We first apply these analyses to standard two-population and randomly-connected networks. We then specifically exploit these results to build low-rank connectivity matrices that implement specific trajectories to transiently encode specified stimuli, and examine the noise-robustness and capacity of this setup.

## Results

We study linear networks of $N$ randomly and recurrently coupled rate units with dynamics given by:

$$\dot{r}_i = -r_i + \sum_{j=1}^{N} J_{ij} r_j + I(t) r_{0,i}. \tag{1}$$

Such networks can be interpreted as describing the linearized dynamics of a system around an equilibrium state. In this picture, the quantity $r_i$ represents the deviation of the activity of the unit $i$ from its equilibrium value. For simplicity, in the following we refer to the quantity $r_i$ as the firing rate of unit $i$. Here $J_{ij}$ denotes the effective strength of the connection from neuron $j$ to neuron $i$. Unless otherwise specified, we consider an arbitrary connectivity matrix **J**. Along with the recurrent input, each unit $i$ receives an external drive $I(t)r_{0,i}$ in which the temporal component $I(t)$ is equal for all neurons, and the vector $\mathbf{r}_0$ (normalized to unity) represents the relative amount of input to each neuron.

### Monotonic vs. amplified transient trajectories

We focus on the transient autonomous dynamics in the network following a brief input in time ($I(t) = \delta(t)$) along the external input direction $\mathbf{r}_0$, which is equivalent to setting the initial condition to $\mathbf{r}_0$. The temporal activity of the network in response to this input can be represented as a trajectory $\mathbf{r}(t)$ in the high-dimensional space in which the $i$-th component is the firing rate of neuron $i$ at time $t$. We assume the network is stable, so that the trajectory asymptotically decays to the equilibrium state that corresponds to $r_i = 0$. At intermediate times, depending on the connectivity matrix **J** and on the initial condition $\mathbf{r}_0$, the trajectory can however exhibit two qualitatively different types of behavior: it can either monotonically decay towards the asymptotic state or transiently move away from it (Fig 1A and 1B). We call these two types of trajectories respectively monotonic and amplified.

The two types of transient trajectories can be distinguished by looking at the Euclidean distance between the activity at time point $t$ and the asymptotic equilibrium state, given by the activity norm $\| \mathbf{r}(t) \| = \sqrt{r_1(t)^2 + r_2(t)^2 + \ldots + r_N(t)^2}$. Focusing on the norm allows us to deal with a single scalar quantity instead of $N$ firing rates. Monotonic and amplified transient trajectories respectively correspond to monotonically decaying and transiently increasing $\| \mathbf{r}(t) \|$ (Fig 1C). Note that a transiently increasing $\| \mathbf{r}(t) \|$ necessarily implies that the firing rate of at least one neuron shows a transient increase in its absolute value before decaying to zero.

One approach to understanding how the connectivity matrix **J** determines the transient trajectory is to project the dynamics on the basis formed by the right-eigenvectors $\{\mathbf{v}_k\}$ of **J** [36]. The component $\tilde{r}_k(t)$ along the $k$—th eigenmode decays exponentially and the activity norm can be expressed as:

$$\| \mathbf{r}(t) \| = \sqrt{\sum_{k=1}^{N} \tilde{r}_k(t)^2 + 2\sum_{k>j} \tilde{r}_k(t)\tilde{r}_j(t)(\mathbf{v}_k \cdot \mathbf{v}_j)}. \tag{2}$$

If all the eigenvectors $\mathbf{v}_k$ are mutually orthogonal, then the squared activity norm is a sum of squares of decaying exponentials, and therefore a monotonically decaying function. Connectivity matrices **J** with all orthogonal eigenvectors are called normal matrices, and they thus generate only monotonic transients. In particular, any symmetric matrix is normal. On the other hand, connectivity matrices for which some eigenvectors are not mutually orthogonal are called *non-normal* [37]. For such matrices, the second term under the square root in Eq (2) can have positive or negative sign, so that the norm cannot in general be written as the sum of decaying exponentials. It is well known that non-normal matrices can lead to non-monotonic transient trajectories [28, 31–35, 38].

Nonetheless, a non-normal connectivity matrix **J** is just a necessary, but not a sufficient condition for the existence of transiently amplified trajectories. As will be illustrated below,

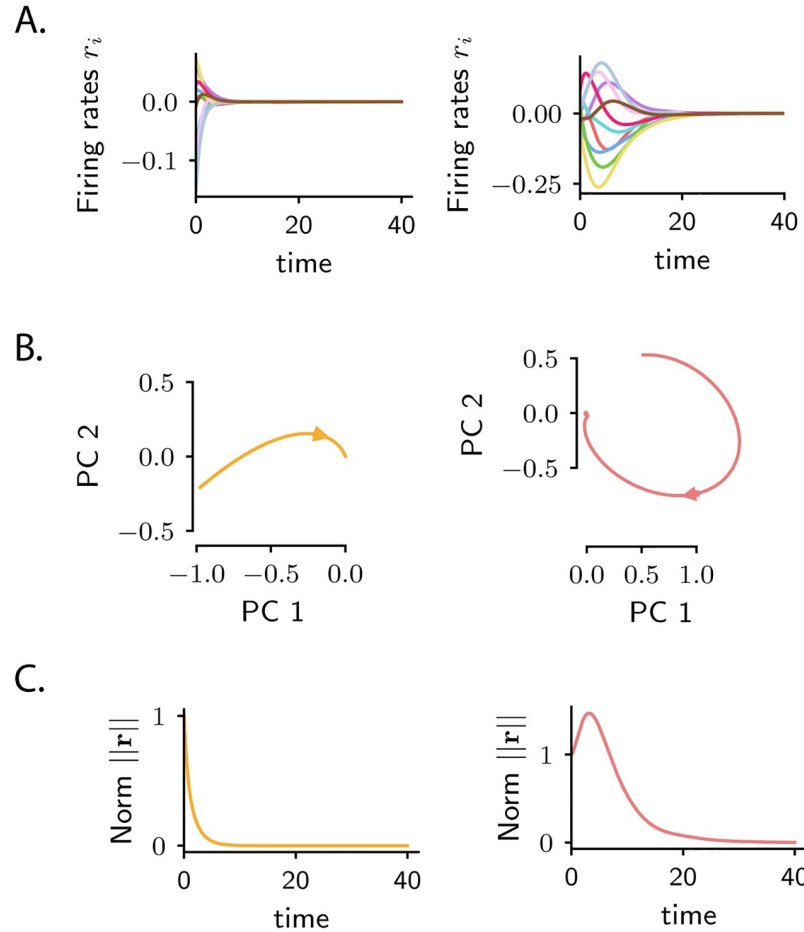

**Fig 1. Monotonically decaying vs. amplified transient dynamics.** Dynamics of a linear recurrent network in response to a short external perturbation along a given input direction $\mathbf{r}_0$. The left and right examples correspond to two different connectivity matrices, where the connection strengths are independently drawn from a Gaussian distribution with zero mean and variance equal to $g^2/N$ (left: $g = 0.5$; right: $g = 0.9$). **A.** Firing rate dynamics of 10 individual units. **B.** Projections of the population activity onto the first two principal components of the dynamics. Yellow and red color correspond respectively to $g = 0.5$ and $g = 0.9$. **C.** Temporal dynamics of the activity norm $\|\mathbf{r}(t)\|$. *Left*: in the case of weakly non-normal connectivity the activity norm displays monotonic decaying behaviour for any external input perturbation. *Right*: for strongly non-normal connectivity, specific stimuli generate a transient increase of the activity norm. $N = 200$ in simulations.

having non-orthogonal eigenvectors does not guarantee the existence of transiently amplified inputs. This raises the question of identifying the sufficient conditions on the connectivity matrix $\mathbf{J}$ and input $\mathbf{r}_0$ for the transient trajectory to be amplified. In the following, we point out a simple criterion on the connectivity matrix $\mathbf{J}$ for the existence of amplified trajectories, and show that it is possible to identify the input subspace giving rise to amplified trajectories and estimate its dimensionality.

## Two classes of non-normal connectivity

To distinguish between monotonic and amplified trajectories, we focus on the rate of change $d\|\mathbf{r}(t)\|/dt$ of the activity norm. For a monotonic trajectory, this rate of change is negative at all

times, while for amplified trajectories it transiently takes positive values before becoming negative as the activity decays to the equilibrium value. Using this criterion, we can determine the conditions under which a network generates an amplified trajectory for at at least one input $\mathbf{r}_0$. Indeed, the rate of change of the activity norm satisfies (see [38, 39])

$$\frac{1}{\|\mathbf{r}\|}\frac{\mathrm{d}\|\mathbf{r}\|}{\mathrm{d}t} = \frac{\mathbf{r}^T(\mathbf{J}_S - \mathbf{I})\mathbf{r}}{\|\mathbf{r}\|^2}, \qquad \mathbf{J}_S = \frac{\mathbf{J} + \mathbf{J}^T}{2} \tag{3}$$

Here the matrix $\mathbf{J}_S$ denotes the symmetric part of the connectivity matrix $\mathbf{J}$. The right hand side of Eq (3) is a Rayleigh quotient [40]. It reaches its maximum value when $\mathbf{r}(t)$ is aligned with the eigenvector of $\mathbf{J}_S$ associated with its largest eigenvalue, $\lambda_{\max}(\mathbf{J}_S)$, and the corresponding maximal rate of change of the activity norm is therefore $\lambda_{\max}(\mathbf{J}_S) - 1$.

Eq (3) directly implies that a necessary and sufficient condition for the existence of transiently amplified trajectories is that the largest eigenvalue of the symmetric part $\mathbf{J}_S$ be larger than unity, $\lambda_{\max}(\mathbf{J}_S) > 1$ [38]. If that is the case, choosing the initial condition along the eigenvector associated with $\lambda_{\max}(\mathbf{J}_S)$ leads to a positive rate of change of the activity norm at time $t = 0$, and therefore generates a transient increase of the norm corresponding to an amplified trajectory, which shows the sufficiency of the criterion. Conversely, if a given input produces an amplified trajectory, at least one eigenvalue of $\mathbf{J}_S$ is necessarily larger than one. If that were not the case, the right hand side of the equation for the norm would take negative values for all vectors $\mathbf{r}(t)$, implying a monotonic decay of the norm. This demonstrates the necessity of the criterion.

The criterion based on the symmetric part of the connectivity matrix allows us to distinguish two classes of connectivity matrices: if $\lambda_{\max}(\mathbf{J}_S) < 1$ all external inputs $\mathbf{r}_0$ lead to monotonically decaying trajectories (non-amplifying connectivity); if $\lambda_{\max}(\mathbf{J}_S) < 1$ specific input directions lead to a non-monotonic amplified activity norm (amplifying connectivity). The key point here is that for a non-normal connectivity matrix $\mathbf{J}$, the symmetric part $\mathbf{J}_S$ is in general different from $\mathbf{J}$. The condition for the stability of the system ($\mathfrak{Re}\lambda_{\max}(\mathbf{J}) < 1$) and the condition for transient amplification ($\lambda_{\max}(\mathbf{J}_S) > 1$) are therefore not mutually exclusive. This is instead the case for normal networks, which include symmetric, anti-symmetric, orthogonal connectivity matrices and trivial one-dimensional dynamics.

The simplest illustration of this result is a two-population network. In that case the relationship between the eigenvalues of $\mathbf{J}$ and $\mathbf{J}_S$ is straightforward. The eigenvalues of $\mathbf{J}$ and $\mathbf{J}_S$ are given by

$$\lambda^{\pm}(\mathbf{J}) = \frac{\mathrm{Tr}(\mathbf{J}) \pm \sqrt{\mathrm{Tr}^2(\mathbf{J}) - 4\mathrm{Det}(\mathbf{J})}}{2}, \qquad \lambda^{\pm}(\mathbf{J}_S) = \frac{\mathrm{Tr}(\mathbf{J}) \pm \sqrt{\mathrm{Tr}^2(\mathbf{J}) - 4\mathrm{Det}(\mathbf{J}) + 4\Delta^2}}{2}, \tag{4}$$

where $\mathrm{Tr}(\mathbf{J})$ and $\mathrm{Det}(\mathbf{J})$ are the trace and determinant of the full connectivity matrix $\mathbf{J}$, and $2\Delta$ is the difference between the off-diagonal elements of $\mathbf{J}$. Assuming for simplicity that the eigenvalues of $\mathbf{J}$ are real, Eq (4) show that the maximal eigenvalue of $\mathbf{J}_S$ is in general larger than the maximal eigenvalue of $\mathbf{J}$, and the difference between the two is controlled by the parameter $\Delta$ which quantifies how non-symmetric the matrix $\mathbf{J}$ is. If $\Delta$ is large enough, $\mathbf{J}_S$ will have an unstable eigenvalue, even if both eigenvalues of $\mathbf{J}$ are stable (Fig 2A). The value of $\Delta$ therefore allows to distinguish between non-amplifying and amplifying connectivity. Furthermore, for amplifying connectivity, the parameter $\Delta$ directly controls the amount of amplification in the network (Fig 2B), defined as the maximum value of the norm $\|\mathbf{r}(t)\|$ over time and initial conditions $\mathbf{r}_0$ (see Methods). A specific example is a network consisting of an excitatory and an inhibitory population [32]. In that case our criterion states that the excitatory feedback needs to be

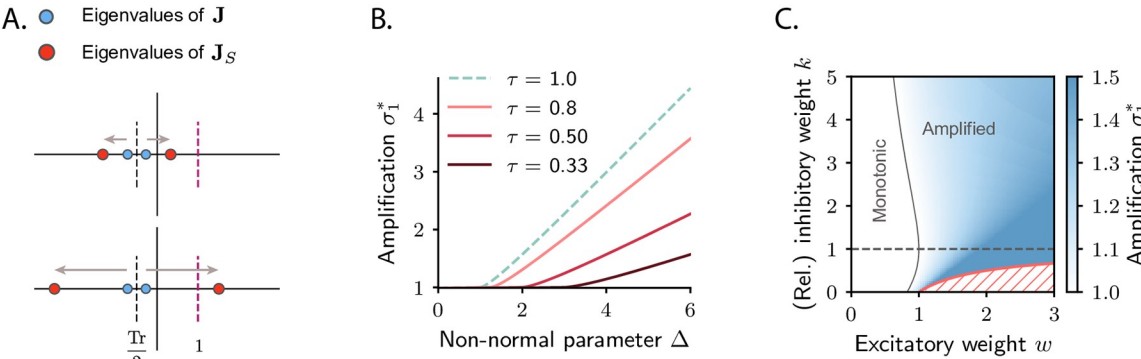

**Fig 2. Dynamical regimes for a network of two interacting populations. A**. Relation between the eigenvalues of the connectivity matrix **J** (blue dots) and the eigenvalues of its symmetric part, **$J_S$** (red dots). Both pairs of eigenvalues are symmetrically centered around Tr(**J**)/2, but the eigenvalues of **$J_S$** lie further apart (Eq 4), and the maximal eigenvalue of **$J_S$** can cross unity if the difference 2Δ between the off-diagonal elements of the connectivity matrix is sufficiently large (bottom panel). **B**. Value of the maximum amplification of the system (quantified by the maximal singular value $\sigma_1(\mathbf{P}_{t^*})$ of the propagator, see Methods) as a function of the non-normal parameter Δ. Here we fix the two eigenvalues of **J**, the largest of which effectively determines the largest timescale of the dynamics, and vary Δ. Colored traces correspond to different values of the largest timescale of the system $\tau = 1/(1 - \Re\lambda_{\max}(\mathbf{J}))$. For small values of Δ the maximum amplification is equal to one, and it increases approximately linearly when Δ is larger than the critical value. Each colored trace corresponds to a different choice of Tr(**J**) and Det(**J**). From top to bottom traces: Tr(**J**) = 0, −0.5, −2, −4 and Det(**J**) = Tr²(**J**)/4 (for convenience), corresponding respectively to $\tau = 1, 0.8, 0.5, 0.33$. The trace for $\tau = 1$ corresponds to $\Re\lambda_{\max}(\mathbf{J}) = 0$. **C**. Dynamical regimes for an excitatory-inhibitory two population model, as in [32]. Here $w$ represents the weights of the excitatory connections ($J_{EE} = J_{IE} = w$) and $-kw$ the weights of the inhibitory ones ($J_{EI} = J_{II} = -kw$). The inhibition-dominated regime corresponds to $k > 1$. The color code corresponds to the maximum amplification, as quantified by the maximal singular value $\sigma_1(\mathbf{P}_{t^*})$. The grey trace corresponds to the boundary between the monotonic and the amplified parameter regions. The red trace represents the stability boundary, with the unstable region hatched in red. In order to achieve transient amplification the excitatory weight $w$ has to be approximately larger than unity, when $k$ is larger than but close to one. Note that amplification can be obtained also for $0 < k < 1$, in a parameter region limited by the stability boundary.

(approximately) larger than unity in order to achieve transient amplification, when $k$ is larger than but close to one (Fig 2C and S6 Text).

A second illustrative example is a network of $N$ randomly connected neurons, where each connection strength is independently drawn from a Gaussian distribution with zero mean and variance equal to $g^2/N$. For such a network, the eigenvalues of **J** and **$J_S$** are random, but their distributions are known. The eigenvalues of **J** are uniformly distributed in the complex plane on a circle of radius $g$ [41], so that the system is stable for $g < 1$ (Fig 3A). On the other hand, the eigenvalues of the symmetric part **$J_S$** are real and distributed according to the semicircle law with spectral radius $\sqrt{2}g$ [42, 43] (Fig 3B). The fact that the spectral radius of **$J_S$** is larger by a factor $\sqrt{2}$ than the spectral radius of **J** implies that if $g$ is in the interval $1/\sqrt{2} < g < 1$ the network is stable but exhibits amplified transient activity (Fig 3C). Note that the connectivity is non-normal for any value of $g$, but the additional condition $g > 1/\sqrt{2}$ is needed for the existence of amplified trajectories. This in particular implies that for random connectivity transient amplification requires the network to be close to instability, so that the dynamics are slowed down as pointed out in [34].

## Coding with amplified transients

For a connectivity matrix satisfying the amplification condition $\lambda_{\max}(\mathbf{J}_S) > 1$, only specific external inputs $\mathbf{r}_0$ are amplified by the recurrent circuitry, while others lead to monotonically decaying trajectories (Fig 4B). Which and how many orthogonal inputs are amplified? What is the resulting state of the network at the time of maximal amplification, and how can the inputs be decoded from that state?

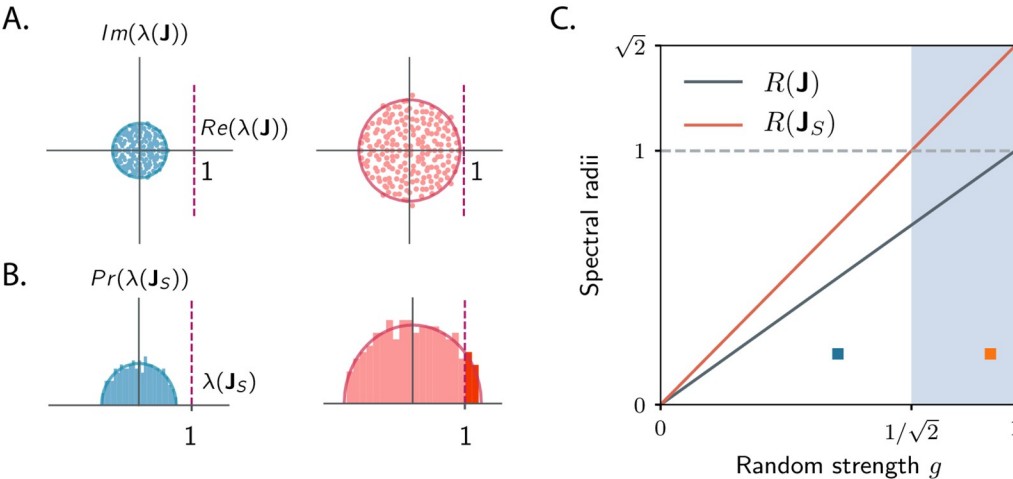

**Fig 3. Dynamical regimes of a *N*-dimensional network model with random Gaussian connectivity structure.** Each entry of **J** is independently drawn from a Gaussian distribution with zero mean and variance $g^2/N$. **A.** The eigenvalues of **J** are complex, and, in the limit of large *N*, distributed uniformly within a circle of radius $R(\mathbf{J}) = g$ in the complex plane (Girko's law, [41]). The system is stable if $g < 1$. Left: $g = 0.5$. Right: $g = 0.9$. **B.** The eigenvalues of the symmetric part $\mathbf{J}_S$ are real-valued, and are distributed in the large *N* limit according to the semicircle law, with the largest eigenvalue of $\mathbf{J}_S$ given by the spectral radius $R(\mathbf{J}_S) = \sqrt{2}g$ [42, 43]. Since the spectral radius of $\mathbf{J}_S$ is larger than the spectral radius of **J**, for sufficiently large values of *g* some eigenvalues of $\mathbf{J}_S$ can be larger than unity (in red), while the network dynamics are stable ($g < 1$). **C.** Spectral radii of **J** and $\mathbf{J}_S$ as a function of the random strength *g*. The interval of values of *g* for which the system displays strong transient dynamics in response to specific inputs is given by $1/\sqrt{2} < g < 1$. $N = 200$ in simulations.

One approach to these questions is to examine the mapping from inputs to states at a given time *t* during the dynamics. Since we consider linear networks, the state reached at time *t* from the initial condition $\mathbf{r}_0$ is given by the linear mapping $\mathbf{r}(t) = \mathbf{P}_t\,\mathbf{r}_0$, where for any time $t > 0$, $\mathbf{P}_t = \exp(t(\mathbf{J} - \mathbf{I}))$ is an $N \times N$ matrix called the propagator of the network. At a given time *t*, the singular value decomposition (SVD) of $\mathbf{P}_t$ defines a set of singular values $\{\sigma_k^{(t)}\}$, and two sets of orthonormal vectors $\{\mathbf{R}_k^{(t)}\}$ and $\{\mathbf{L}_k^{(t)}\}$, such that $\mathbf{P}_t$ maps $\mathbf{R}_k^{(t)}$ onto $\sigma_k^{(t)}\mathbf{L}_k^{(t)}$. In other words, taking $\mathbf{R}_k^{(t)}$ as the initial condition leads the network to the state $\sigma_k^{(t)}\mathbf{L}_k^{(t)}$ at time *t*:

$$\mathbf{r}(t) = \mathbf{P}_t\mathbf{R}_k^{(t)} = \sigma_k^{(t)}\mathbf{L}_k^{(t)}. \tag{5}$$

If $\sigma_k^{(t)} > 1$, the norm of the activity at time *t* is larger than unity, so that the initial condition $\mathbf{R}_k^{(t)}$ is amplified. In fact, the largest singular value of $\mathbf{P}_t$ determines the maximal possible amplification at time *t* (see Methods). Note that for a normal matrix, the left and right singular vectors $\mathbf{R}_k^{(t)}$ and $\mathbf{L}_k^{(t)}$ are identical, and the singular values are equal to the modulus of the eigenvalues, so that the stability of the dynamics imply an absence of amplification. Conversely, stable amplification implies that $\mathbf{R}_k^{(t)}$ and $\mathbf{L}_k^{(t)}$ are not identical, so that an amplified trajectory explores at least two dimensions corresponding to the plane spanned by $\mathbf{R}_k^{(t)}$ and $\mathbf{L}_k^{(t)}$.

Since the propagator $\mathbf{P}_t$ depends on time, the singular vectors $\mathbf{R}_k^{(t)}$ and $\mathbf{L}_k^{(t)}$, and the singular values $\sigma_k^{(t)}$ depend on time. One can therefore look at the temporal trajectories $\sigma_k^{(t)}$, which by definition all start at one at $t = 0$ (Fig 4A). If the connectivity satisfies the condition for transient amplification, at least one singular value increases above unity, and reaches a maximum before asymptotically decreasing to zero. The number of singular values that simultaneously take values above unity (Fig 4A) defines the number of orthogonal initial conditions amplified by the dynamics. Choosing a time $t^*$ at which $N_s$ of the singular value trajectories lie above

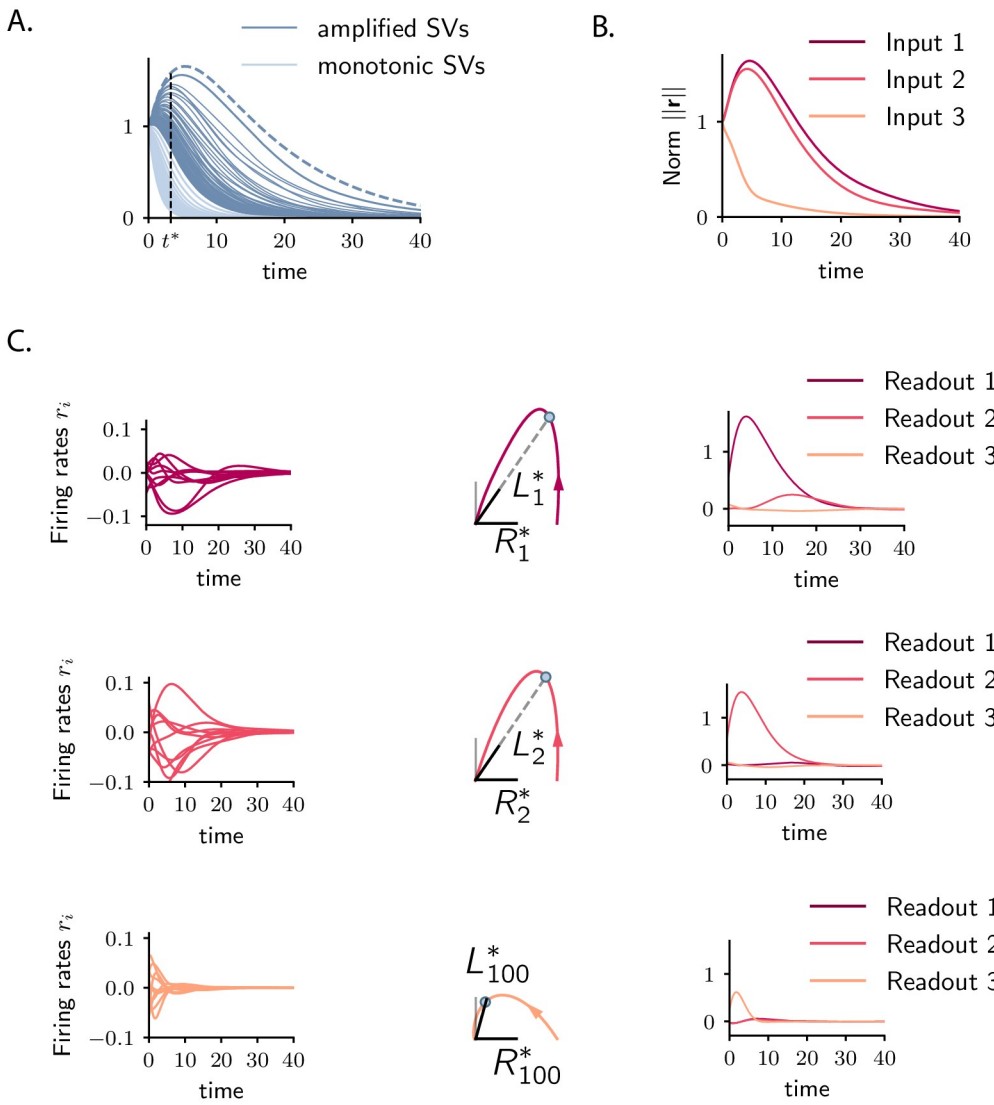

**Fig 4. Coding multiple stimuli with amplified transient trajectories.** Example corresponding to a *N*-dimensional Gaussian connectivity matrix with *g* = 0.9. **A**. Singular values of the propagator, $\sigma_i^{(t)}$, as a function of time (SV trajectories). Dark blue traces show the amplified singular values, defined as having positive slope at time *t* = 0; The dominant singular value $\sigma_1^{(t)}$ corresponds to the dashed line. Light blue traces correspond to the non-amplified singular values, having negative slope at *t* = 0. **B**. Norm of the activity elicited by the first two amplified inputs, i.e. $\mathbf{R}_1^*$, $\mathbf{R}_2^*$, (right singular vectors corresponding to singular values $\sigma_1^{(t^*)}$ and $\sigma_2^{(t^*)}$ at time *t** indicated by the dashed vertical line in panel A; purple and red traces), and by one non-amplified input (chosen as $\mathbf{R}_{100}^*$, corresponding to $\sigma_{100}^{(t)}$; orange trace). **C**. Illustration of the dynamics elicited by the three inputs, $\mathbf{R}_1$, $\mathbf{R}_2$ and $\mathbf{R}_{100}$ (shown in different rows), as in **B**. *Left*: Activity of 10 individual units. *Center*: Projections of the evoked trajectories onto the plane defined by the stimulus $\mathbf{R}_i^*$ and the corresponding readout vector $\mathbf{L}_i^*$ (in analogy with the amplified case, we chose the readout of the non-amplified dynamics to be the state of the system at time *t**, i.e. $\mathbf{L}_{100}^*$). *Right*: population responses to the three stimuli projected on the readout vectors $\mathbf{L}_1^*$, $\mathbf{L}_2^*$ and $\mathbf{L}_{100}^*$. *N* = 1000 in simulations.

unity, we can indeed identify a set of $N_s$ orthogonal, amplified inputs corresponding to the right singular vectors $\mathbf{R}_k^{(t^*)}$ of the propagator at time *t**. According to Eq (5), each of these inputs is mapped in an amplified fashion to the corresponding left singular vector $\mathbf{L}_k^{(t^*)}$ at time *t**, which also form an orthogonal set. Each amplified input can therefore be decoded by

projecting the network activity on the corresponding left singular vector $\mathbf{L}_k^{(t^*)}$ (Fig 4C). Since $\{\mathbf{L}_k^{(t)}\}$ are mutually orthogonal, the different initial conditions lead to independent encoding channels. Again, as the dynamics are non-normal, the inputs $\mathbf{R}_k$ and the outputs $\mathbf{L}_k$ are not identical, so that the dynamics for each amplified input are at least two-dimensional (Fig 4C).

How many independent, orthogonal inputs can a network encode with amplified transients? To estimate this number, a central observation is that the slopes of the different singular value trajectories at $t = 0$ are given by the eigenvalues of the symmetric part of the connectivity $\mathbf{J}_S$. This follows from the fact that the singular values of the propagator $\mathbf{P}_t$ are the square root of the eigenvalues of $\mathbf{P}_t^T \mathbf{P}_t$, and at short times $\delta t$ we have $\mathbf{P}_{\delta t}^T \mathbf{P}_{\delta t} \simeq \mathbf{I} + 2(\mathbf{J}_S - \mathbf{I})\delta t$. This implies that the number of singular values with positive slope at the initial time is equal to the number of eigenvalues of the symmetric part $\mathbf{J}_S$ larger than unity. To eliminate the trajectories with small initial slopes, one can further constrain the slopes to be larger than a margin $\epsilon$, in which case the number of amplified trajectories $N_S(\epsilon)$ is given by the number of eigenvalues of $\mathbf{J}_S$ larger than $1 + \epsilon$. Note that $N_S(\epsilon)$ provides only a lower bound on the number of amplified inputs, as singular values with initial slope smaller than zero can increase at later times. It is straightforward to compute $N_S(\epsilon)$ when the connectivity matrix $\mathbf{J}$ is a random matrix with independent and identically distributed elements. In this case the probability distribution of the eigenvalues of its symmetric part $\mathbf{J}_S$ follows the semicircle law (Fig 3), and when the number of neurons $N$ is large, the number $N_s$ of amplified inputs scales linearly with $N$.

To summarize, the amplified inputs and the corresponding encoding at peak amplification can be determined directly from the singular value decomposition of the propagator, given by the exponential of the connectivity matrix. For an arbitrary $N \times N$ matrix $\mathbf{J}$, characterizing analytically the SVD of its exponential is in general a complex and to our knowledge open mathematical problem. For specific classes of matrices, the propagator and its SVD can however be explicitly computed, and in the following we will exploit this approach.

## Implementing specific transient trajectories

The approach outlined above holds for any arbitrary connectivity matrix, and allows us to identify the external inputs which are strongly amplified by the recurrent structure, along with the modes that get most activated during the elicited transients, and therefore encode the inputs. We now turn to the converse question: how to choose the network connectivity $\mathbf{J}$ such that it generates a pre-determined transient trajectory. Specifically, we focus on low-rank networks, a type of model ubiquitous in neuroscience [44, 45], and set out to determine the minimal-rank connectivity that transiently transforms a fixed, arbitrary input $\mathbf{r}_0$ into a fixed, arbitrary output $\mathbf{w}$ at the time of peak amplification, through two-dimensional dynamics.

To address this question, we consider a connectivity structure given by a unit-rank matrix $\mathbf{J} = \Delta \mathbf{u}\mathbf{v}^T$ [45]. Here $\mathbf{u}$ and $\mathbf{v}$ are two vectors with unitary norm and correlation $\rho$ ($\langle \mathbf{u}, \mathbf{v} \rangle = \rho$), and $\Delta$ is an overall scaling parameter. We applied to this connectivity the general analysis outlined above (see Methods). The only non-zero eigenvalue of $\mathbf{J}$ is $\Delta\rho$, and the corresponding linear system is stable for $\Delta\rho < 1$. The largest eigenvalue of the symmetric part of the connectivity $\mathbf{J}_S$ is given by $\Delta(\rho + 1)/2$, so that the network displays amplified transients if and only if $\Delta(\rho + 1)/2 > 1$ (while $\Delta\rho < 1$). Keeping the eigenvalue $\Delta\rho$ constant and increasing $\Delta$ will therefore lead to a transition from monotonically decaying to amplified transients (Fig 5A). If $\rho = 0$, the vectors $\mathbf{u}$ and $\mathbf{v}$ are orthogonal, and the condition for amplification is simply $\Delta > 2$. Note that in this situation, amplification is obtained without slowing down the dynamics, in contrast to randomly coupled networks [34].

For this unit rank connectivity matrix, the full propagator $\mathbf{P}_t = \exp(\mathbf{t}(\mathbf{J} - \mathbf{I}))$ of the dynamics can be explicitly computed (see Methods). The non-trivial dynamics are two-dimensional, and

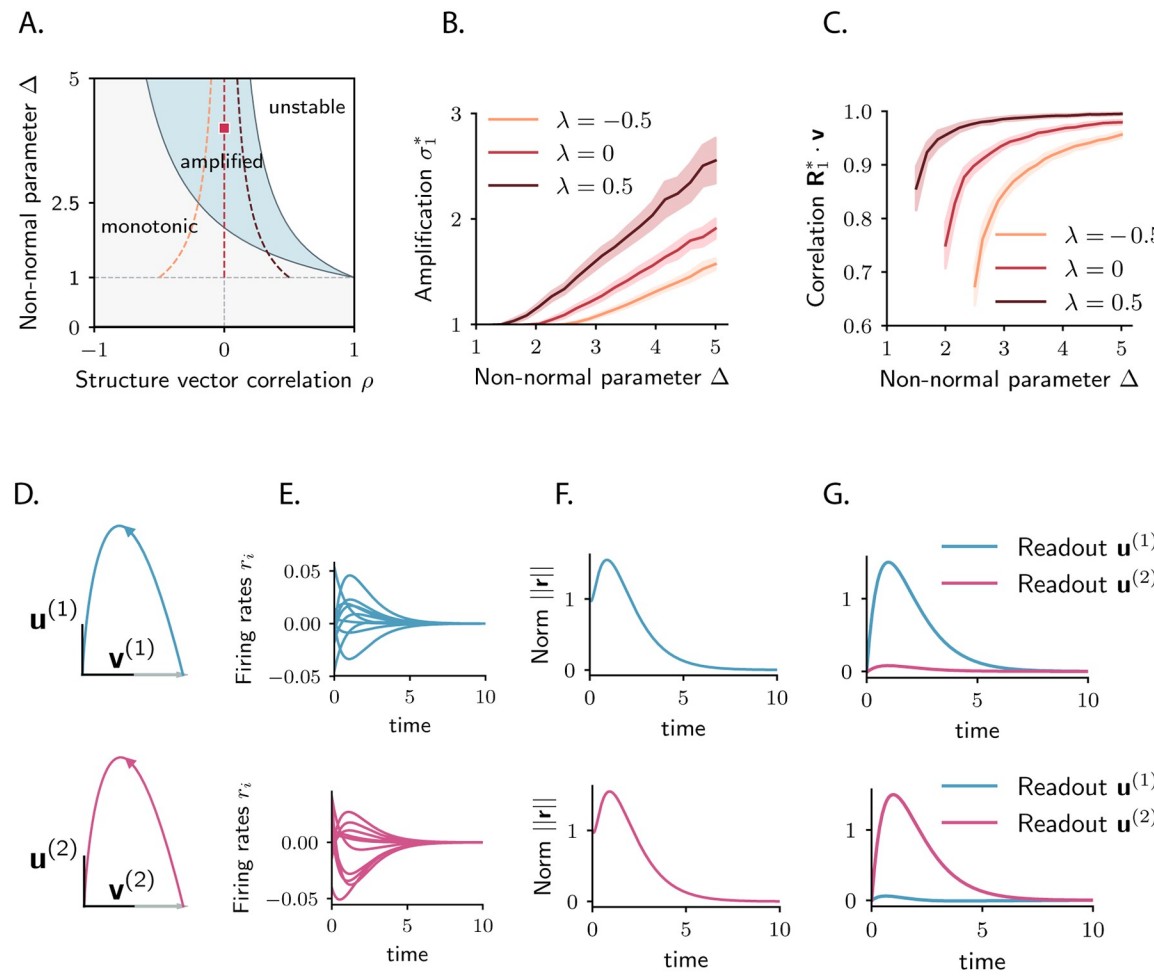

**Fig 5. Low-dimensional amplified dynamics in random networks with unit-rank structure. A**. Dynamical regimes as a function of the structure vector correlation $\rho = \mathbf{u} \cdot \mathbf{v}$ and the scaling parameter of the connectivity matrix, $\Delta$. Grey shaded areas correspond to parameter regions where the network activity is monotonic for all inputs; blue shaded areas indicate parameter regions where the network activity is amplified for specific inputs; for parameter values in the white area, activity is unstable. Samples of dynamics are shown in the bottom panels, for parameter values indicated by the colored dot in the phase diagram: $\Delta = 4$ and $\rho = 0$. Dashed colored traces correspond to the parameter regions explored in panels **B**. and **C**., defined by the equation $\lambda = \Delta\rho$. **B**. Maximum amplification of the system, quantified by $\sigma_1(\mathbf{P}_{t^*})$, the first singular value of the propagator, as a function of the scaling parameter $\Delta$. Here we fix the eigenvalue of the connectivity matrix $\lambda = \Delta\rho$ associated with the eigenvector $\mathbf{u}$, and vary $\Delta$. Colored traces correspond to different choices of the eigenvalue of the connectivity $\lambda$. **C**. Correlation between the optimally amplified input direction $\mathbf{R}_1^*$ and the structure vector $\mathbf{v}$ as a function of the parameter $\Delta$. Increasing the non-normal parameter $\Delta$ aligns the optimally amplified input with the structure vector $\mathbf{v}$. In **B**. and **C**. mean and standard deviation over 50 realizations of the connectivity matrix are shown for each trace. The elements of the structure vectors are drawn from a Gaussian distribution, so that they have on average unit norm and correlation $\rho$ (see Methods). **D**. Low-dimensional dynamics in the case of two stored patterns. Input $\mathbf{v}^{(1)}$ (resp. $\mathbf{v}^{(2)}$) elicits a two-dimensional trajectory which brings the activity along the other structure vector $\mathbf{u}^{(1)}$ (resp. $\mathbf{u}^{(2)}$), mapping stimulus $\mathbf{v}^{(1)}$ (resp. $\mathbf{v}^{(2)}$) into its transient readout $\mathbf{u}^{(1)}$ (resp. $\mathbf{u}^{(2)}$). Blue and red colors correspond to the two stored patterns. **E**. Firing rates of 10 individual units. **F**. Temporal evolution of the activity norm. **G**. Projection of the network response evoked by the input along $\mathbf{v}^{(1)}$ (resp. $\mathbf{v}^{(2)}$) on the corresponding readout $\mathbf{u}^{(1)}$ (resp. $\mathbf{u}^{(2)}$). The case of unit rank connectivity (one stored pattern) reduces to the first row of panels **D**. –**G**. (where the activity on $\mathbf{u}^{(2)}$ is equivalent to the activity on a readout orthogonal to $\mathbf{u}^{(1)}$). $N = 3000$ in simulations.

lie in the plane spanned by the structure vectors $\mathbf{u}$ and $\mathbf{v}$ (Fig 5D), while all components orthogonal to this plane decay exponentially to zero. Determining the singular value decomposition of the propagator allows us to compute the amount of amplification of the system, as the value of $\sigma_1(\mathbf{P}_t)$ at the time of its maximum $t^*$. In the amplified regime (for $\Delta(\rho + 1)/2 > 1$), the amount of amplification increases monotonically with $\Delta$ (Fig 5B). Since only one eigenvalue of

$\mathbf{J}_S$ is larger than unity, only one input perturbation is able to generate amplified dynamics. For large values of $\Delta$, this optimal input direction is strongly correlated with the structure vector $\mathbf{v}$. Perturbing along the vector $\mathbf{v}$ elicits a two-dimensional trajectory which at its peak amplification is strongly correlated with the other structure vector $\mathbf{u}$ (Fig 5C). Choosing $\mathbf{v} = \mathbf{r}_0$ and $\mathbf{u} = \mathbf{w}$, the unit-rank connectivity therefore directly implements a trajectory that maps the input $\mathbf{r}_0$ into the output $\mathbf{w}$, identified as the transient readout vector for stimulus $\mathbf{r}_0$.

Several, orthogonal trajectories can be implemented by adding orthogonal unit-rank components. For instance, taking $\mathbf{J} = \Delta \mathbf{u}^{(1)} \mathbf{v}^{(1)T} + \Delta \mathbf{u}^{(2)} \mathbf{v}^{(2)T}$, where the planes defined by the structure vectors in each term are mutually orthogonal, the input $\mathbf{v}^{(1)}$ evokes a trajectory which is confined to the plane defined by $\mathbf{u}^{(1)}$ and $\mathbf{v}^{(1)}$, and which maps the input $\mathbf{v}^{(1)}$ into the output $\mathbf{u}^{(1)}$ at the time of peak amplification (Fig 5D–5G). Similarly, the input $\mathbf{v}^{(2)}$ is mapped into the output $\mathbf{u}^{(2)}$ during the evoked transient dynamics. Therefore, the rank-2 connectivity $\mathbf{J}$ implements two transient patterns, encoding the stimuli $\mathbf{v}^{(1)}$ and $\mathbf{v}^{(2)}$ into the readouts $\mathbf{u}^{(1)}$ and $\mathbf{u}^{(2)}$. A natural question is how robust the scheme is and how many patterns can be implemented in a network of fixed size $N$.

## Robustness and capacity

To investigate the robustness of the transient coding scheme implemented with unit-rank terms, we first examined the effect of additional random components in the connectivity. Adding to each connection a random term of variance $g^2/N$ introduces fluctuations of order $g\Delta^2/\sqrt{N}$ to the component of the activity on the plane defined by $\mathbf{u}$ and $\mathbf{v}$ (see Methods). Consequently, the projection of the trajectory on the readout $\mathbf{w} = \mathbf{u}$ has fluctuations of the same order (Fig 6A–6C). A supplementary effect of random connectivity is to add to the dynamics a component orthogonal to $\mathbf{u}$ and $\mathbf{v}$, proportional to $\Delta$ (see S8 Text), which however does not contribute to the readout along $\mathbf{w}$. Thus, for large $N$, the randomness in the synaptic connectivity does not impair the decoding of the stimulus $\mathbf{r}_0$ from the activity along the corresponding readout $\mathbf{w}$.

The robustness of the readouts to random connectivity implies in particular that the unit-rank coding scheme is robust when an extensive number $P$ of orthogonal transient trajectories are implemented by the connectivity $\mathbf{J}$. To show this, we generalize the unit-rank approach and consider a rank-$P$ connectivity matrix, given by the sum of $P$ unit-rank matrices, $\mathbf{J} = \Delta \sum_{p=1}^{P} \mathbf{u}^{(p)} \mathbf{v}^{(p)T}$, where each term specifies an input-output pair. We focus on the case where the elements of all the vectors $\mathbf{u}^{(p)}$ and $\mathbf{v}^{(p)}$ are independently drawn from a random distribution (see Methods), implying that all input-output pairs are mutually orthogonal, i.e. uncorrelated, in the limit of large $N$. In this situation, the interaction between the dynamics evoked by one arbitrary input $\mathbf{v}^{(p)}$ and the additional $P - 1$ patterns is effectively described by a system with connectivity $\mathbf{J} = \Delta \mathbf{u}^{(p)} \mathbf{v}^{(p)T}$ corrupted by a random component with zero mean and variance equal to $\Delta^2 P/N^2$ (see Methods). From the previous results, it follows that the fluctuations of the activity of the readout $\mathbf{u}^{(p)}$ are of order $\Delta^3 \sqrt{P}/N$ (Fig 6D–6F). Thus, in high dimension, the readout activity is robust to the interactions between multiple encoded trajectories. When the number of encoded trajectories is extensive ($P = O(N)$), each stimulus $\mathbf{v}^{(p)}$, can therefore still be decoded from the projection of the activity on the corresponding readout $\mathbf{u}^{(p)}$.

A natural upper bound on the number of trajectories that can be implemented by the connectivity $\mathbf{J}$ is derived from the stability constraints of the linear system. Indeed, the largest eigenvalues of $\mathbf{J}$ is given by $\Delta\sqrt{P/N}$ and it needs to be smaller than one for stability. Thus, the maximum number of trajectories that can be encoded in the connectivity $\mathbf{J}$ is given by $P_{\max} = N/\Delta^2$ and defines the capacity of the network. Crucially, the capacity scales linearly with

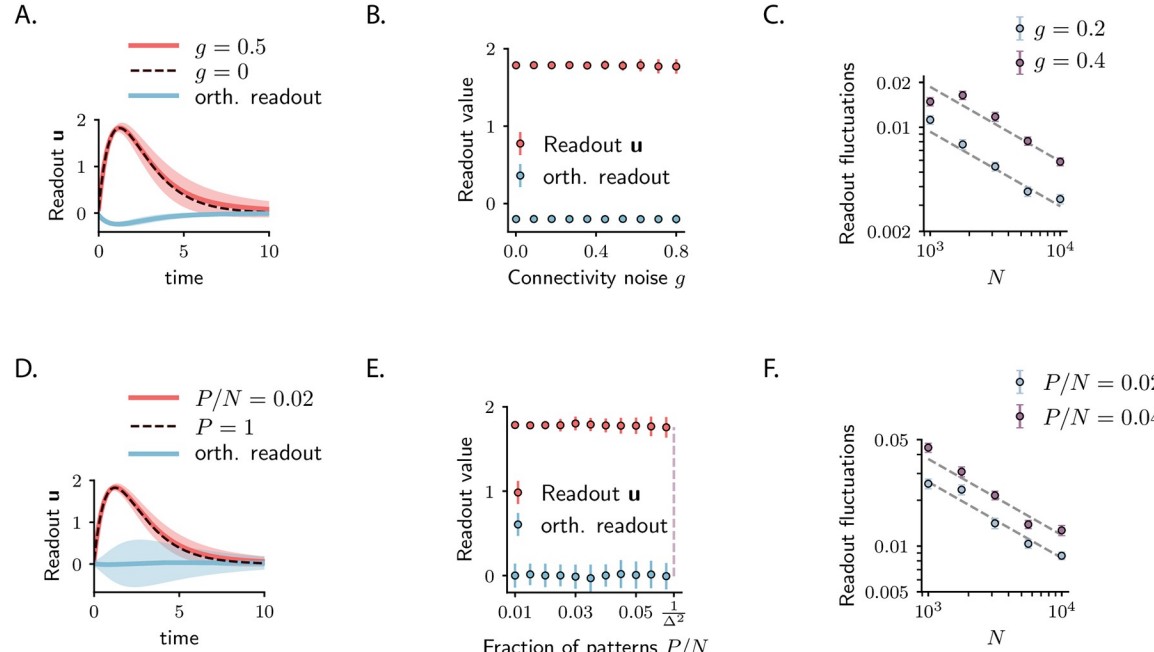

**Fig 6. Robustness of the transient coding scheme and capacity of the network. (A-B-C)** Robustness of the readout activity for a single stored pattern **u-v** in presence of randomness in the connectivity with variance $g^2/N$. **A**. Projection of the population activity elicited by input **v** along the readout **u** (red trace) and along a readout orthogonal to **u** (blue trace) for $g = 0.5$. The elements of the orthogonal readout are drawn from a random distribution with mean zero and variance $1/N$ and are fixed over trials. The projection of the activity on **u** is also shown for the zero noise case ($g = 0$; black dashed line). **B**. Value of the activity along **u** (red dots) and along the orthogonal readout (blue dots) at the peak amplification ($t = t^*$), as a function of $g$. In **A** and **B**, $N = 200$; error bars correspond to the standard deviation over 100 realizations of the random connectivity. **C**. Standard deviation of the readout activity at the peak amplification as a function of the network size $N$ for two values of $g$. The fluctuations are inversely proportional to the network size and scale as $g\Delta^2/\sqrt{N}$. Error bars correspond to the standard deviation of the mean over 100 realization of the connectivity noise. **(D-E-F)** Robustness of the transient coding scheme in presence of multiple stored patterns. **D**. Projection of the population activity elicited by one arbitrary amplified input $\mathbf{v}^{(k)}$ along the corresponding readout $\mathbf{u}^{(k)}$ (red trace) and along a different arbitrary readout $\mathbf{u}^{(k')}$ (blue trace) for $P/N = 0.02$. The readout $\mathbf{u}^{(k')}$ was changed for every trial. The projection of the activity on $\mathbf{u}^{(k)}$ is also shown when only the pattern $\mathbf{u}^{(k)}$-$\mathbf{v}^{(k)}$ is encoded ($P = 1$; black dashed line). **E**. Value of the activity along $\mathbf{u}^{(k)}$ (red dots) and along the readout $\mathbf{u}^{(k)}$ (blue dots) at the peak amplification ($t = t^*$), as a function of $P/N$. In **D** and **E** $N = 200$; error bars correspond to the standard deviation over 100 realizations of the connectivity matrix. **F**. Standard deviation of the readout activity (along $\mathbf{u}^{(k)}$) at the peak amplification as a function of the network size $N$ for two values of $P/N$. The fluctuations are inversely proportional to the network size and scale as $\Delta^3\sqrt{P}/N$. Error bars correspond to the standard deviation of the mean over 100 realizations of the connectivity noise.

the size of the network $N$. The capacity also decreases for highly amplified systems, resulting in a trade-off between the separability of the neural activity evoked by different stimuli (quantified by $\Delta$) and the number of stimuli that can be encoded in the connectivity (quantified by $P_{max}$).

## Discussion

We examined the conditions under which linear recurrent networks can implement an encoding of stimuli in terms of amplified transient trajectories. The fundamental mechanism underlying amplified transients relies on the non-normal properties of the connectivity matrix, i.e. the fact that the left- and right-eigenvectors of the connectivity matrix are not identical [37]. A number of recent studies in theoretical neuroscience have pointed out the interesting dynamical properties of networks with non-normal connectivity [28, 31–35, 46, 47]. Several of these works [28, 32, 34, 35] have examined the amplification of the norm of the activity vector, as we do here. However, it was not pointed out that the presence of amplification can be diagnosed by considering the eigenvalues of the symmetric part $\mathbf{J}_S$ of the connectivity matrix (rather than

examining properties of the eigenvectors of the connectivity matrix **J**), leading to the distinction of two classes of recurrent networks. This general criterion appears to be well-known in the theory of linear systems (Theorem 17.1 in [37]). Here we applied it to standard models of recurrent networks used in computational neuroscience, and in particular to low-rank networks [45].

We have shown that the largest eigenvalue of the symmetric part of the connectivity defines the amplification properties of the system asymptotically at small times $t$. Yet, it does not provide a direct measure of the maximum amplification of the norm $\|\mathbf{r}(t)\|$ over all times $t$ (see Maximum amplification of the system). The maximum amplification can be derived using the singular value decomposition (SVD) of the propagator $\mathbf{P}_t$, which however can be computed analytically only for simple connectivity matrices. To quantify the maximum amplification other measures have been developed that rely on the so-called pseudospectra [37] of the connectivity matrix, a generalization of the eigenvalue spectra useful for the study of non-normal dynamics. While the spectrum of the symmetric part of the connectivity controls the amplification of the system at small times and the eigenvalue spectrum determines its large time dynamics, the transient dynamics at intermediate times is largely determined by the properties of the pseudospectra of the connectivity **J** (Chapter 4 of [37]). Notably, the result known as the Kreiss Matrix Theorem (Eq. 14.8 and 14.12 in [37]) provides a lower and upper bound for the maximum amplification of $\|\mathbf{r}(t)\|$ based on the pseudospectrum of the connectivity **J**.

Applying the criterion for transient amplification to classical randomly connected networks, we found that amplification occurs only in a narrow parameter region close to the instability, where the dynamics substantially slow down as previously shown [34]. To circumvent this issue, and produce strong transient amplification away from the instability, [28] introduced stability-optimized circuits (SOCs) in which inhibition is fine-tuned to closely balance excitation, and demonstrated that such dynamics can account for the experimental data recorded in the motor cortex [24]. We showed here that low-rank networks can achieve the same purpose, and exhibit strong, fast amplification in a large parameter region away from the instability. One difference with SOCs is that low-rank networks explicitly implement low-dimensional dynamics that transform a specified initial state into a specified, orthogonal output state. Several low-rank channels could be combined to reproduce higher-dimensional dynamics similar to those observed during the generation of complex movements [24].

In our framework we modeled the external stimulus as the initial condition of the network, and the amplified dynamics is autonomously generated by the recurrent interactions. Although this might appear as an oversimplifying assumption, it has nevertheless been proven useful to describe the transient population activity in motor and sensory areas. In motor and pre-motor cortex, the initial condition of the population dynamics during the execution of the movement may be set by the phase of preparatory neural activity that precedes the movement, and may determine to a large extent the time course of the movement-related dynamics [22–24]. A similar mechanism has been recently proposed to underlie the generation and population coding properties of strong sensory responses following stimulus offsets in auditory cortex. Here different auditory stimuli result in largely orthogonal initial conditions at the stimulus offset, thus generating orthogonal population offset responses across stimuli [48]. The assumption of autonomous dynamics does not hold when naturalistic (e.g. temporally structured) stimuli are considered [49]. Understanding how more complex external inputs are transformed by the non-normal amplified network dynamics constitutes a major direction of future work.

The study by Murphy and Miller [32] reported that the excitatory-inhibitory (EI) structure of cortical networks induces non-normal amplification between so-called sum and difference E-I modes. Interestingly, the specific networks they considered are of the low-rank type, with

sum and difference modes corresponding to left- and right- vectors of the individual unit-rank terms [35]. This connectivity structure is therefore a particular instance of the low-rank implementation of amplified trajectories that we described here. Moreover, Murphy and Miller specifically focused on the inhibition-dominated regime [50], which as we show approximately corresponds to the class of unit-rank E-I networks that exhibit strong transient amplification (Fig 2 and Supp Info; note that these networks can exhibit amplification also for $0 < k \leq 1$, in a parameter region limited by the stability boundary). In the present study, we have not enforced a separation between excitatory and inhibitory neurons, but this can be done in a straightforward way by adding a unit-rank term in which all excitatory (resp. inhibitory) connections have the same weight, and these weights are chosen strong enough to make all excitatory (resp. inhibitory) synapses positive (resp. negative). This additional component would induce one more amplified channel that would correspond to the global E-I difference mode of Murphy and Miller.

Here our aim was to produce amplified, but not necessarily long-lasting transients. The timescale of the transients generated using the unit-rank implementation is in fact determined by the effective timescale of the network, set by the dominant eigenvalue of the connectivity matrix. As shown in previous studies that focused on implementing transient memory traces [31, 33, 46], longer transients can be obtained either by increasing recurrent feedback (i.e. the overlap between vectors in the unit-rank implementation), or by creating longer hidden feed-forward chains. For instance, an effective feed-forward chain of length $k$ can be obtained from a rank $k$ connectivity term of the type $\mathbf{J} = \Delta\mathbf{v}^{(k+1)}\mathbf{v}^{(k)T} + \ldots + \Delta\mathbf{v}^{(3)}\mathbf{v}^{(2)T} + \Delta\mathbf{v}^{(2)}\mathbf{v}^{(1)T}$, i.e. in which each term feeds into the next one [51]. This leads in general to a $k + 1$-dimensional transient with a timescale extended by a factor $k$ [33]. Implementing this kind of higher-dimensional transients naturally comes at the cost of reducing the corresponding capacity of the network.

The implementation of transient channels proposed here clearly bears a strong analogy with Hopfield networks [44]. The aim of Hopfield networks is to store patterns of activity in memory as fixed points of the dynamics, and this is achieved by adding to the connectivity matrix a unit-rank term $\boldsymbol{\xi}\boldsymbol{\xi}^T$ for each pattern $\boldsymbol{\xi}$. One key difference with the present network is that Hopfield networks rely on symmetric connectivity [52], while amplified transients are obtained by using strongly asymmetric terms in which the left- and right-vectors are possibly orthogonal. Another difference is that Hopfield networks rely on a non-linearity to generate fixed points for each pattern, while here we considered instead linear dynamics in the vicinity of a single fixed-point. The non-linearity of Hopfield networks endows them with error-correcting properties, in the sense that a noisy initial condition will always lead to the activation of a single memorized pattern. A weaker form of error-correction is also present in our linear, transient encoding, since any component along non-amplified directions will decay faster than the amplified pattern. However, if two amplified patterns are simultaneously activated, they will lead to the activation of both corresponding outputs. This absence of competition may not be undesirable, as it can allow for the simultaneous encoding, and possibly binding, of several complementary stimulus features.

The amplified dynamics map specific external inputs onto orthogonal patterns of activity with larger values of the norm $\|\mathbf{r}(t)\|$. These dynamics however, along with amplifying the amplitude of the signal, also amplify the external noise that is injected in the network. This external noise is maximally amplified along the readout dimensions corresponding to the amplified inputs (Eq (83)), implying that the signal-to-noise ratio at the peak amplification is comparable to the SNR at the initial state. Therefore, transient amplification may not favor stimulus decoding during the transient state with respect to the initial state in presence of noisy input, but may nonetheless be needed to keep a stable value of the SNR across the transient dynamics (see Robustness of the readout activity). Instead, the amplification of the norm

constitutes an advantage when the synaptic connections to the readout neurons are themselves corrupted by noise. As the noise in the readout weights, or observational noise, is not directly fed into the network, it does not get amplified by the recurrent interactions. As a result, the detrimental effects of observational noise are overcome by amplifying the signal $\mathbf{r}(t)$ above the noise level, which can be directly implemented by the transient coding scheme illustrated here. Transient amplification of external inputs may therefore result in an increased ability to robustly decode the external input in presence of noisy readout synapses.

While we focused here on linear dynamics in the vicinity of a fixed point, strong non-linearities can give rise to different transient phenomena [12]. In particular, one prominent proposal is that robust transient coding can be implemented using stable heteroclinic channels, i.e. sequences of saddle points that feed into each other [4]. This mechanism has been exploited in specific models based on clustered networks [5]. A general theory for this type of transient coding is to our knowledge currently lacking, and constitutes an interesting avenue for future work.

## Methods

### The network model

We study a recurrent network of $N$ randomly coupled rate units. Each unit $i$ is described by the time-dependent variable $r_i(t)$, representing its firing rate at time $t$. The transfer function of the individual units is linear, so that the equation governing the temporal dynamics of the network reads:

$$\tau \dot{r}_i = -r_i + \sum_{j=1}^{N} J_{ij} r_j + I(t) r_{0,i}, \tag{6}$$

where $\tau$ represents the membrane time constant (fixed to unity), and $J_{ij}$ is the effective synaptic strength from neuron $j$ to neuron $i$. In absence of external input, the system has only one fixed point corresponding to $r_i = 0$ for all $i$. To have stable dynamics, we require that the eigenvalues of the connectivity matrix $\mathbf{J}$ be smaller than unity, i.e. $\mathfrak{Re}\lambda_{\max}(J) < 1$. We write the external input as the product between a common time-varying component $I(t)$, and a term $r_{0,i}$ which corresponds to the relative activation of each unit. The terms $r_{0,i}$ can be arranged in a $N$-dimensional vector $\mathbf{r}_0$, which we call the external input direction. Here we focus on very short external input durations ($I(t) = \delta(t)$) and on input directions of unit norm ($\|\mathbf{r}_0\| = 1$). This type of input is equivalent to setting the initial condition to $\mathbf{r}(0) = \mathbf{r}_0$. Since we study a linear system, varying the norm of the input direction would result in a linear scaling of the dynamics.

### Dynamics of the network

We first outline the standard approach to the dynamics of the linear network defined by Eq (6) (see e.g. [36, 53]). The solution of the differential equation given by Eq (6) can be obtained by diagonalizing the linear system, i.e. by using a change of basis $\mathbf{r} = \mathbf{V}\tilde{\mathbf{r}}$ such that the connectivity matrix in the new basis $\mathbf{\Lambda} = \mathbf{V}^{-1}\mathbf{J}\mathbf{V}$ is diagonal. The matrix $\mathbf{V}$ contains the eigenvectors $\mathbf{v}_1, \mathbf{v}_2, \ldots, \mathbf{v}_N$ of the connectivity $\mathbf{J}$ as columns, while $\mathbf{\Lambda}$ has the corresponding eigenvalues $\lambda_i$ on the diagonal. Therefore the variables $\tilde{\mathbf{r}}$ represent the components of the rate vector on the basis of eigenvectors of $\mathbf{J}$. In this new basis the system of coupled equations in Eq (6) reduces to the set of uncoupled equations

$$\dot{\tilde{r}}_i = -\tilde{r}_i + \lambda_i \tilde{r}_i + \delta(t)\tilde{r}_{0,i}. \tag{7}$$

The dynamics of the linear network given by Eq (6) can thus be written in terms of its

components on the eigenvectors $\mathbf{v}_i$ as

$$\mathbf{r}(t) = \sum_{i=1}^{N} \tilde{r}_i(t)\mathbf{v}_i, \qquad \tilde{r}_i(t) = e^{t(\lambda_i - 1)/} \tilde{r}_{0,i}. \tag{8}$$

Equivalently, the solution of the linear system can be expressed as the product between a linear, time-dependent operator $\mathbf{P}_t$ and the initial condition $\mathbf{r}_0$ [54]:

$$\mathbf{r}(t) = \mathbf{P}_t \, \mathbf{r}_0. \tag{9}$$

The linear operator $\mathbf{P}_t$ is called the propagator of the system and it is defined as the matrix exponential of the connectivity matrix $\mathbf{J}$, i.e. $\mathbf{P}_t = \exp(t(\mathbf{J} - \mathbf{I})/\tau)$. By using the definition of matrix exponential in terms of power series, we can express the propagator as $\mathbf{P}_t = \mathbf{V} \, \mathbf{diag}(e^{t(\lambda_1 - 1)}, \ldots, e^{t(\lambda_N - 1)})\mathbf{V}^{-1}$. From Eq (9) we note that the propagator $\mathbf{P}_t$ at time $t$ defines a mapping from the state of the system at time $t = 0$, i.e. the external input direction $\mathbf{r}_0$, to the state $\mathbf{r}(t)$.

## Dynamics of the norm

To study the amplification properties of the network, we follow [39] and focus on the temporal dynamics of the population activity norm $\|\mathbf{r}(t)\|$ [28]. The equation governing the dynamics of the norm can be derived by writing $\|\mathbf{r}\| = \sqrt{\mathbf{r}^T\mathbf{r}}$, so that the relative rate of change of the norm is given by [39]

$$\begin{aligned} \frac{1}{\|\mathbf{r}\|} \frac{\mathrm{d}\|\mathbf{r}\|}{\mathrm{d}t} &= \frac{1}{\sqrt{\mathbf{r}^T\mathbf{r}}} \frac{\mathrm{d}\sqrt{\mathbf{r}^T\mathbf{r}}}{\mathrm{d}t} \\ &= \frac{1}{2\mathbf{r}^T\mathbf{r}} \left( \frac{\mathrm{d}\mathbf{r}^T}{\mathrm{d}t}\mathbf{r} + \mathbf{r}^T\frac{\mathrm{d}\mathbf{r}}{\mathrm{d}t} \right). \end{aligned} \tag{10}$$

By using Eq (6) we can write the right hand side of the previous equation as

$$\begin{aligned} \frac{1}{\|\mathbf{r}\|} \frac{\mathrm{d}\|\mathbf{r}\|}{\mathrm{d}t} &= \frac{\mathbf{r}^T((\mathbf{J}^T - \mathbf{I}) + (\mathbf{J} - \mathbf{I}))\mathbf{r}}{2\|\mathbf{r}\|^2} \\ &= \frac{\mathbf{r}^T(\mathbf{J}_S - \mathbf{I})\mathbf{r}}{\|\mathbf{r}\|^2}, \end{aligned} \tag{11}$$

where we introduced $\mathbf{J}_S = (\mathbf{J} + \mathbf{J}^T)/2$, the symmetric part of the connectivity matrix $\mathbf{J}$.

Both the eigenvalues and the eigenvectors of $\mathbf{J}_S$ provide information on the transient dynamics of the system. On one hand, we show in the main text that the activity norm can have non-monotonic behaviour if and only if at least one eigenvalue of the matrix $\mathbf{J}_S$ is larger than one. Therefore the eigenvalues of $\mathbf{J}_S$ determine the type of transient regime of the system. On the other hand, as $\mathbf{J}_S$ is symmetric, its set of eigenvectors is orthogonal and provides a useful orthonormal basis onto which we can project the dynamics. In this basis, the connectivity matrix is given by $\mathbf{J}' = \mathbf{V}_S^T\mathbf{J}\mathbf{V}_S$, where $\mathbf{V}_S$ contains the eigenvectors of $\mathbf{J}_S$ as columns. The matrix $\mathbf{J}$ can be uniquely decomposed as $\mathbf{J} = \mathbf{J}_S + \mathbf{J}_A$, where $\mathbf{J}_A = (\mathbf{J} - \mathbf{J}^T)/2$ is the anti-symmetric part of $\mathbf{J}$, so that

$$\mathbf{J}' = \mathbf{diag}(\lambda_1(\mathbf{J}_S), \ldots, \lambda_N(\mathbf{J}_S)) + \mathbf{V}_S^T\mathbf{J}_A\mathbf{V}_S. \tag{12}$$

The first term on the right hand side is a diagonal matrix, while the second term is an anti-symmetric matrix. Since the latter has zero diagonal elements, the new connectivity matrix $\mathbf{J}'$

displays the eigenvalues of $\mathbf{J}_S$ on the diagonal. The off-diagonal terms of $\mathbf{J}'$ are given by the elements of $\mathbf{V}_S^T \mathbf{J}_A \mathbf{V}_S$ and represent the strength of the couplings between the eigenvectors of $\mathbf{J}_S$. In the amplified regime, some of the eigenvalues of $\mathbf{J}_S$ are larger than one, so that without the coupling between the modes of $\mathbf{J}_S$, the connectivity $\mathbf{J}'$ would be unstable. However, in our case $\mathbf{J}$ and $\mathbf{J}'$ are stable matrices, meaning that the coupling terms ensure the stability of the overall system. Moreover, varying the strengths of the coupling terms while keeping fixed the diagonal terms affects in a non-trivial way the maximum amplification of the system. Therefore, the decomposition in Eq (12) allows us to identify the set of key parameters that controls the maximum amplification of a specific system. In the following, we will systematically use this decomposition to analyze specific classes of matrices.

## Amplification

To identify which inputs are amplified, we examine the dynamics of the activity norm $\|\mathbf{r}(t)\|$ for an arbitrary external input $\mathbf{r}_0$. The one-dimensional Eq (11) alone is not enough to determine the time course of $\|\mathbf{r}(t)\|$, since the right hand side depends on the solution of the $N$—dimensional system Eq (6). Therefore, for a specific input $\mathbf{r}_0$, we can use Eq (9) and write the norm of the elicited trajectory as

$$\| \mathbf{r}(t) \| = \| \mathbf{P}_t \mathbf{r}_0 \| . \tag{13}$$

**Input-output mapping between amplified inputs and readouts.** The dynamics elicited in response to an input along an arbitrary direction is in general complex. However, the singular value decomposition (SVD) of the propagator provides a useful way to understand the network dynamics during the transient phase. Any matrix $\mathbf{A}$ can be written as

$$\mathbf{A} = \mathbf{L}\Sigma\mathbf{R}^T, \tag{14}$$

where the matrix $\Sigma$ contains the singular values $\sigma_i(\mathbf{A})$ on the diagonal, while the columns of $\mathbf{L}$ (resp. $\mathbf{R}$) are the left (resp. right) singular vectors of $\mathbf{A}$, i.e. the eigenvectors of $\mathbf{A}\mathbf{A}^T$ (resp. $\mathbf{A}^T\mathbf{A}$). The matrices $\mathbf{R}$ and $\mathbf{L}$ are unitary, meaning that they separately provide two orthogonal sets of unitary vectors. Thus, we can write the SVD of the propagator as

$$\mathbf{P}_t = \sigma_1^{(t)}\mathbf{L}_1^{(t)}\mathbf{R}_1^{(t)T} + \sigma_2^{(t)}\mathbf{L}_2^{(t)}\mathbf{R}_2^{(t)T} + \ldots + \sigma_N^{(t)}\mathbf{L}_N^{(t)}\mathbf{R}_N^{(t)T}. \tag{15}$$

From Eq (15) we see that, at a given time $t$, the propagator $\mathbf{P}_t$ maps each right singular vector $\mathbf{R}_k^{(t)}$ into the left singular vector $\mathbf{L}_k^{(t)}$, scaled by the singular value $\sigma_k^{(t)}$ (see Eq (5)). Note that for normal systems the singular value decomposition and the eigen-decomposition coincide. In this case the matrices $\mathbf{L}$ and $\mathbf{R}$ both contain the eigenvectors of $\mathbf{P}_t$ as columns, so that $\mathbf{L}_k^{(t)}$ and $\mathbf{R}_k^{(t)}$ lie on a single dimension. Instead, for a non-normal system the right and left singular vectors do not align along one direction, and the dynamics of the system in response to an input along $\mathbf{R}_k^{(t)}$ spans at least the two dimensions defined by the two vectors $\mathbf{R}_k^{(t)}$ and $\mathbf{L}_k^{(t)}$. The vectors $\mathbf{R}_k^{(t)}$ for which $\sigma_k^{(t)} > 1$ correspond to the amplified inputs at time $t$, while the outputs $\mathbf{L}_k^{(t)}$ are the corresponding readouts at time $t$.

**Number of amplified inputs.** The number of amplified inputs at time $t$ is given by the number of singular values $\sigma_k^{(t)}$ larger than unity. To estimate this number, we examine the temporal dynamics of the singular values $\sigma_k^{(t)}$ in time (SV trajectories). We observe that, for a system in the amplified regime ($\lambda_{\max}(\mathbf{J}_S) > 1$), at least one of the SV trajectories has non-monotonic dynamics, starting from one at $t = 0$ and then increasing before decaying to zero. In fact, the singular values of the propagator at small times $t = \delta t$ are defined as the square

roots of the eigenvalues of

$$\mathbf{P}_{\delta t}^{T}\mathbf{P}_{\delta t} = \mathbf{I} + 2(\mathbf{J}_S - I)\delta t + O(\delta t^2). \tag{16}$$

From Eq (16) we can compute the singular values of $\mathbf{P}_{\delta t}$ as

$$\sigma_k(\mathbf{P}_{\delta t}) = 1 + (\lambda_k(\mathbf{J}_S) - 1)\delta t + O(\delta t^2), \tag{17}$$

so that the slope at time $t = 0$ of the $k$-th singular value of the propagator is

$$\left.\frac{\mathrm{d}\sigma_k}{\mathrm{d}t}\right|_{t=0} = \lambda_k(\mathbf{J}_S) - 1. \tag{18}$$

Eq (18) shows that the number of singular values larger than unity at small times is given by the number of the eigenvalues of $\mathbf{J}_S$ larger than unity, which we denote as $N_S$.

**Maximum amplification of the system.** From Eq (15) we see that the maximum over initial conditions of the amplification at time $t$ corresponds to the dominant singular value of the propagator, $\sigma_1^{(t)}$. The associated amplified input and corresponding readout are respectively $\mathbf{R}_1^{(t)}$ and $\mathbf{L}_1^{(t)}$. To obtain the maximum amplification of the system over inputs and over time, we need to compute the time $t^*$ at which $\sigma_1^{(t)}$ attains its maximum value. Therefore, the value $\sigma_1^{(t^*)}$ quantifies the maximum amplification over inputs and over time, while $\mathbf{R}_1^{(t^*)}$ and $\mathbf{L}_1^{(t^*)}$ correspond respectively to the most amplified input direction and the associated readout.

Interestingly, it can be shown that the input $\mathbf{R}_1^{(t^*)} \equiv \mathbf{R}_1^*$ satisfies the equation (see S1 Text)

$$\mathbf{R}_1^{*T}(\mathbf{J}_S - \mathbf{I})\mathbf{R}_1^* = 0, \tag{19}$$

which depends only on the symmetric part of the connectivity matrix $\mathbf{J}_S$. We will exploit this equation to identify the amplified initial condition $\mathbf{R}_1^*$ in specific cases. Note that, except for $N = 2$, Eq (19) does not fully specify the maximally amplified input.

## Characterizing transient dynamics—Summary

Summarizing, our approach for characterizing its transient dynamics can be divided into three main steps:

1. Compute $\mathbf{J}_S$, along with its eigenvalues and eigenvectors.

2. Compute the propagator of the system $\mathbf{P}_t$.

3. Compute the Singular Value Decomposition (SVD) of the propagator.

These three steps can be in principle performed numerically for any connectivity matrix. For particular classes of connectivity matrices, we show below that some or all three steps are analytically tractable.

## Random Gaussian network

Here we consider a non-normal random connectivity matrix with synaptic strength independently drawn from a Gaussian distribution

$$J_{ij} \sim \mathcal{N}(0, g^2/N). \tag{20}$$

The eigenvalues of $\mathbf{J}$ are complex and uniformly distributed in a circle of radius $g$ [41]:

$$P(\lambda) = \begin{cases} \dfrac{1}{\pi g^2}, & |\lambda| \leq g \\ 0, & |\lambda| > g \end{cases} \tag{21}$$

For this class of matrices, we can analytically determine the condition for amplified transients, and estimate the number of amplified inputs. In the stable regime ($g < 1$), the symmetric part of the connectivity $\mathbf{J}_S$ can have unstable eigenvalues. In fact, the elements of the symmetric part are distributed according to

$$J_{S,ij} \sim \begin{cases} \mathcal{N}(0, g^2/2N), & i \neq j \\ \mathcal{N}(0, g^2/N), & i = j \end{cases} \tag{22}$$

From random matrix theory we know that the eigenvalues of the matrix given by Eq (22) are real and distributed according to the semicircle law [42, 43]:

$$P(\lambda) = \begin{cases} \dfrac{1}{\pi g^2}\sqrt{2g^2 - \lambda^2}, & |\lambda| \leq \sqrt{2}g \\ 0, & |\lambda| > \sqrt{2}g \end{cases} \tag{23}$$

In particular, the spectral radius of $\mathbf{J}_S$ is $\sqrt{2}g$, meaning that $\mathbf{J}_S$ has unstable eigenvalues if $1/\sqrt{2} < g < 1$.

To estimate the number of amplified initial conditions, we compute the lower bound on their number $N_S(\epsilon)$, i.e. the number of eigenvalues of $\mathbf{J}_S$ larger than $1 + \epsilon$:

$$\frac{N_S(\epsilon, g)}{N} = \int_{1+\epsilon}^{\sqrt{2}g} P(\lambda(\mathbf{J}_S))\, d\lambda(\mathbf{J}_S)$$

$$= \frac{1}{2} - \frac{1}{2\pi g^2}(1 + \epsilon)\sqrt{2g^2 - (1+\epsilon)^2} - \frac{1}{\pi}\arctan \frac{1 + \epsilon}{\sqrt{2g^2 - (1+\epsilon)^2}}. \tag{24}$$

The number of eigenvalues of $\mathbf{J}_S$ is maximum when $g$ is close to (but smaller than) unity. In this case Eq (24) at the first order in $\epsilon$ translates to

$$\frac{N_S(\epsilon, 1)}{N} = \left(\frac{1}{2} - \frac{1}{2\pi} - \frac{1}{\pi}\arctan(1)\right) - \frac{1}{2\pi}\epsilon \simeq 0.09 - 0.16\epsilon. \tag{25}$$

Therefore, the maximal capacity of a randomly-connected network is therefore around 10%.

Computing the SVD of the exponential of a $N$-dimensional random matrix is to our knowledge an open mathematical problem. Therefore, for an arbitrary random connectivity matrix, the maximal amount of amplification and the amplified initial conditions are accessible only by numerically computing the SVD of $\exp(t(\mathbf{J} - \mathbf{I}))$.

## Two-dimensional system

In this section we consider connectivity matrices describing networks composed of two interacting units of the form

$$\mathbf{J} = \begin{pmatrix} a & b \\ c & d \end{pmatrix}. \tag{26}$$

The eigenvalues of $\mathbf{J}$ determine the stability of the network and can be expressed in terms of its trace and determinant as follows:

$$\lambda^{\pm} = \frac{\mathrm{Tr}(\mathbf{J}) \pm \sqrt{\mathrm{Tr}^2(\mathbf{J}) - 4\mathrm{Det}(\mathbf{J})}}{2}. \tag{27}$$

For the dynamics to be stable, the largest eigenvalue of $\mathbf{J}$ needs to satisfy $\mathfrak{Re}\lambda^+ < 1$, equivalent to the requirement that $\mathrm{Tr}(\mathbf{J} - \mathbf{I}) < 0$ and $\mathrm{Det}(\mathbf{J} - \mathbf{I}) > 0$. Note that if the two eigenvalues $\lambda^{\pm}$ are real, they are symmetrically centered around $\mathrm{Tr}(\mathbf{J})/2$ on the real axis; if they are complex conjugates they have real part equal to $\mathrm{Tr}(\mathbf{J})/2$ and are symmetrically arranged on either side of the real axis.

**Eigenvalues and eigenvectors of $\mathbf{J}_S$.** The condition for transient amplification is determined by the two eigenvalues of $\mathbf{J}_S$, which read:

$$\lambda_S^{\pm} = \frac{\mathrm{Tr}(\mathbf{J}) \pm \sqrt{\mathrm{Tr}^2(\mathbf{J}) - 4\mathrm{Det}(\mathbf{J}) + 4\Delta^2}}{2}, \tag{28}$$

where we introduced the parameter

$$\Delta = \frac{|b - c|}{2}. \tag{29}$$

$\Delta$ represents the difference between the off-diagonal elements of $\mathbf{J}$, and provides a measure of how far from symmetric the connectivity matrix is ($\Delta = 0$ meaning symmetric connectivity). Note that the equation for the eigenvalues of $\mathbf{J}_S$ (Eq 28) differs from the one for the eigenvalues of $\mathbf{J}$ (Eq 27) by the additive term $4\Delta^2$ under the square root. Under the assumption of a stable connectivity $\mathbf{J}$, there exists a critical value for $\Delta$, given by:

$$\Delta_c = \sqrt{1 - \mathrm{Tr}(\mathbf{J}) + \mathrm{Det}(\mathbf{J})} = \sqrt{\mathrm{Det}(\mathbf{J} - \mathbf{I})} \tag{30}$$

above which the rightmost eigenvalue of $\mathbf{J}_S$ is larger than one, meaning that specific inputs are transiently amplified. Note that for a stable $\mathbf{J}$, we have $\mathrm{Det}(\mathbf{J} - \mathbf{I}) > 0$, implying that $\Delta_c$ is real. Thus, $\Delta$ is the crucial parameter which determines the dynamical regime of the system.

**Decomposition on the modes of $\mathbf{J}_S$.** To identify the parameters which determine the maximum amplification of a system, we project the network dynamics onto the orthonormal basis of eigenvectors of $\mathbf{J}_S$. In the new basis the connectivity matrix is given by Eq (12). Interestingly, the non-normal parameter $\Delta$ directly appears in the expression of the anti-symmetric part $\mathbf{J}_A$, so that we obtain

$$\mathbf{J}' = \begin{pmatrix} \lambda_S^+(\Delta) & \Delta \\ -\Delta & \lambda_S^-(\Delta) \end{pmatrix} \tag{31}$$

up to a sign of the off-diagonal elements. From Eq (31) we see that the non-normal parameter $\Delta$, which determines the dynamical regime of the system, also represents the strength of the coupling between the modes of $\mathbf{J}_S$. For $\Delta > \Delta_c$ we have $\lambda_S^+(\Delta) > 1$. Thus, at small times, any component of the dynamics on the first mode of $\mathbf{J}_S$ is amplified by an amount proportional to $\lambda_S^+(\Delta) - 1$. However, at later times, because of the recurrent feedback of strength $\Delta$ between the modes of $\mathbf{J}_S$, the system reaches a finite amount of amplification and relaxes back to the zero fixed point. In the following we examine how the value of $\Delta$ determines the amount of amplification of the system.

**Propagator of the dynamics.** To examine the dependence of the maximum amplification of the system on the parameter $\Delta$ we compute the propagator $\mathbf{P}_t$ and its SVD. A convenient method to compute the exponential of a matrix is provided in [55] (see S2 Text), which we apply to $\mathbf{J}'$ to obtain

$$\exp(t\mathbf{J}') = x_0(t)\mathbf{I} + x_1(t)\mathbf{J}', \tag{32}$$

where the time-dependent functions $x_0(t)$ and $x_1(t)$ are given by

$$x_0(t) = -\frac{\lambda^-}{\lambda^+ - \lambda^-}e^{\lambda^+ t} + \frac{\lambda^+}{\lambda^+ - \lambda^-}e^{\lambda^- t} \tag{33a}$$

$$x_1(t) = \frac{1}{\lambda^+ - \lambda^-}e^{\lambda^+ t} - \frac{1}{\lambda^+ - \lambda^-}e^{\lambda^- t}. \tag{33b}$$

Here $\lambda^+$ and $\lambda^-$ are the eigenvalues of $\mathbf{J}$ (Eq 27).

**SVD of the propagator.** In order to compute the maximum amplification of the system we next compute the largest singular value of the propagator $\sigma_1(\mathbf{P}_t)$ (see S3 Text):

$$\sigma_1(\mathbf{P}_t) = e^{-t}\sqrt{E(t)^2 + H(t)^2} + e^{-t}\sqrt{F(t)^2}, \tag{34}$$

where

$$\begin{cases} E(t) = x_0(t) + x_1(t)(\lambda_S^+ + \lambda_S^-)/2 \\ F(t) = x_1(t)(\lambda_S^+ - \lambda_S^-)/2 \\ H(t) = x_1(t)\Delta \end{cases} \tag{35}$$

**Maximum amplification of the system.** Here we compute the maximal amount of amplification by evaluating the maximum value in time of the amplification envelope $\sigma_1(\mathbf{P}_t)$ (Eq 34), and examine its dependence on the non-normal parameter $\Delta$. In particular we find that, for large values of $\Delta$, this dependence is linear.

To derive this relationship, we note that the combination $\lambda_S^+ - \lambda_S^- = \sqrt{\mathrm{Tr}(\mathbf{J})^2 - 4\mathrm{Det}(\mathbf{J}) + 4\Delta^2}$ depends on $\Delta$, while $\lambda_S^+ + \lambda_S^- = \mathrm{Tr}(\mathbf{J})$ does not. Therefore in Eq (35) only the functions $H(t)$ and $F(t)$ depend on $\Delta$. In the amplified regime $\Delta \gg \Delta_c$, we have that $H(t) \gg E(t)$ for times $t \gg 1/\Delta$ (while for small times $\delta \ll 1/\Delta$ we have $E(\delta t) = 1 + \mathrm{Tr}(\mathbf{J})\delta t/2 \gg \Delta \delta t = H(\delta t)$). In addition, for large values of $\Delta$, we can write $(\lambda_S^+ - \lambda_S^-)/2 = \Delta + O(\Delta^{-1})$ so that the singular value can be written as

$$\sigma_1(\mathbf{P}_t) \simeq e^{-t}(|H(t)| + |F(t)|) \simeq \Delta e^{-t}x_1(t), \qquad \text{for} \qquad t \gg 1/\Delta, \ \Delta \gg \Delta_c. \tag{36}$$

To find the value of the maximum amplification we need to compute the time $t^*$ of occurrence of the global maximum of $\sigma_1(\mathbf{P}_t)$ and the value $\sigma_1(\mathbf{P}_{t^*})$. The final result is given by

$$t^* = \underset{t}{\mathrm{argmax}}\ e^{-t}x_1(t) = \frac{1}{\lambda^+ - \lambda^-}\log\left(\frac{\lambda^- - 1}{\lambda^+ - 1}\right), \tag{37}$$

$$\sigma_1(\mathbf{P}_{t^*}) = \frac{\Delta}{\lambda^+ - \lambda^-}\left[\left(\frac{\lambda^- - 1}{\lambda^+ - 1}\right)^{\frac{\lambda^+}{\lambda^+ - \lambda^-}} - \left(\frac{\lambda^- - 1}{\lambda^+ - 1}\right)^{\frac{\lambda^-}{\lambda^+ - \lambda^-}}\right]. \tag{38}$$

The two-dimensional model given by Eq (26) has four free parameters, namely the strengths of the four recurrent connections. In our analysis we fix the values of the trace $\text{Tr}(\mathbf{J})$ and determinant $\text{Det}(\mathbf{J})$ of the connectivity matrix, so that the dynamics are stable, and vary the parameter $\Delta$. This implies fixing the eigenvalues $\lambda^{\pm}$ and the corresponding timescales $\tau^{\pm} = 1/(1 - \mathfrak{Re}\,\lambda^{\pm})$. This approach allows us to explore how different degrees of symmetry in the connectivity, as quantified by $\Delta$, influence the dynamics while keeping the timescales constant. Thus, we find that, for $\Delta \gg \Delta_c$, and for fixed $\lambda^{\pm}$, the maximum amplification of the system scales linearly with the non-normal parameter $\Delta$.

**Optimally amplified initial condition.** Here we compute the optimal input direction $\mathbf{R}_1^*$ by solving Eq (19). We parametrize the optimal input by the angle $\theta^*$ it forms with the first mode of $\mathbf{J}_S$, i.e. $\mathbf{R}_1^* = (\cos\theta^*, \sin\theta^*)^T$. Thus, Eq (19) translates into

$$\lambda_S^+ \cos^2\theta^* + \lambda_S^- \sin^2\theta^* - 1 = 0 \tag{39}$$

which is satisfied by

$$\theta^* = \pm\arctan\sqrt{\frac{\lambda_S^+ - 1}{1 - \lambda_S^-}}. \tag{40}$$

## Rank-1 connectivity

In this section we consider a unit-rank connectivity matrix defined by

$$\mathbf{J} = \Delta\mathbf{u}\mathbf{v}^T, \tag{41}$$

where the vectors $\mathbf{u}$ and $\mathbf{v}$ are two $N$-dimensional vectors generated as

$$\mathbf{u} = \mathbf{x}_1$$
$$\mathbf{v} = \rho\,\mathbf{x}_1 + \sqrt{1 - \rho^2}\,\mathbf{x}_2,$$

where the vectors $\mathbf{x}_1$, $\mathbf{x}_2$ and $\mathbf{y}$ are $N$-dimensional vectors with components drawn from a Gaussian distribution with mean zero and variance $1/N$ and $\rho$ is a number between $-1$ and $1$ [45]. The average norm and correlation are given by $\langle\mathbf{u}\cdot\mathbf{u}\rangle = \langle\mathbf{v}\cdot\mathbf{v}\rangle = 1$ and $\langle\mathbf{u}\cdot\mathbf{v}\rangle = \rho$, and $\Delta$ is an overall scaling parameter. We consider only positive values of $\Delta$, since a minus sign can be absorbed in the correlation coefficient $\rho$. The matrix $\mathbf{J}$ has $N - 1$ eigenvalues equal to zero and one eigenvalue given by $\lambda = \Delta\rho$, associated with the eigenvector $\mathbf{u}$. In the two-dimensional plane spanned by $\mathbf{u}$ and $\mathbf{v}$, the direction orthogonal to $\mathbf{v}$ specifies another eigenvector of $\mathbf{J}$ corresponding to one of the zero eigenvalues.

**Eigenvalues and eigenvectors of $\mathbf{J}_S$.** We first compute the eigenvalues and eigenvectors of the symmetric part of the connectivity

$$\mathbf{J}_S = \Delta\frac{\mathbf{u}\mathbf{v}^T + \mathbf{v}\mathbf{u}^T}{2}. \tag{42}$$

$\mathbf{J}_S$ is a rank-2 matrix, meaning it has in general two non-zero eigenvalues given by

$$\lambda_S^{\pm} = \frac{\text{Tr}\mathbf{J}_S \pm \sqrt{\left(\text{Tr}\mathbf{J}_S\right)^2 - 4\text{Det}'\mathbf{J}_S}}{2}. \tag{43}$$

Here $\text{Det}'\mathbf{J}_S = \lambda_S^+ \lambda_S^-$ denotes the determinant of $\mathbf{J}_S$ restricted to the $\mathbf{u}\mathbf{v}$-plane, i.e. the determinant of the $2 \times 2$ matrix $[\mathbf{u}, \mathbf{u}_\perp]^T\mathbf{J}_S[\mathbf{u}, \mathbf{u}_\perp]$, where $\mathbf{u}_\perp$ is a vector perpendicular to $\mathbf{u}$ on the $\mathbf{u}\mathbf{v}$-plane (the determinant of the full matrix $\mathbf{J}_S$ is zero because of the zero eigenvalues of $\mathbf{J}_S$). We

find that the two non-zero eigenvalues of the symmetric part $\mathbf{J}_S$ are given by (see S4 Text)

$$\lambda_S^\pm = \frac{\lambda \pm \Delta}{2}. \tag{44}$$

Note that the eigenvalues of $\mathbf{J}_S$ are symmetrically centered around $\lambda/2$, and their displacement is controlled by the scaling parameter $\Delta$. The condition for the system to be in the regime of transient amplification is therefore

$$\frac{\lambda + \Delta}{2} > 1. \tag{45}$$

To compute the eigenvectors $\mathbf{x}_S^\pm$ associated with the non-zero eigenvalues $\lambda_S^\pm$ we have to solve the eigenvector equation

$$(\Delta \mathbf{u}\mathbf{v}^T + \Delta \mathbf{v}\mathbf{u}^T - 2\lambda_S^\pm \mathbf{I})\mathbf{x}_S^\pm = 0. \tag{46}$$

Since the two eigenvectors lie on the $\mathbf{u}\mathbf{v}$-plane, we can write them in the form $\mathbf{x}_S^+ = \mathbf{u} + \alpha\mathbf{v}$ and $\mathbf{x}_S^- = \mathbf{u} + \beta\mathbf{v}$. Solving the eigenvector equation for $\alpha$ and $\beta$ yields $\alpha = 1$ and $\beta = -1$. The two normalized eigenvectors of $\mathbf{J}_S$ are thus given by

$$\mathbf{x}_S^\pm = \frac{\mathbf{u} \pm \mathbf{v}}{\sqrt{2(1 \pm \rho)}}. \tag{47}$$

**Decomposition on the modes of $\mathbf{J}_S$.** We can project the dynamics of the system on the basis of eigenvectors of $\mathbf{J}_S$. Let $\mathbf{V}_S$ be the $N$-dimensional matrix containing the eigenvectors of $\mathbf{J}_S$ as columns:

$$\mathbf{V}_S = (\mathbf{x}_S^+, \mathbf{x}_S^-, \boldsymbol{\xi}_1, \dots, \boldsymbol{\xi}_{N-2}), \tag{48}$$

where the $\boldsymbol{\xi}_i$'s are $N - 2$ arbitrary vectors orthogonal to both $\mathbf{u}$ and $\mathbf{v}$. The projection of the connectivity matrix $\mathbf{J}$ onto the modes of $\mathbf{J}_S$ yields the new connectivity $\mathbf{J}'$:

$$\mathbf{J}' = \mathbf{V}_S^T \mathbf{u}\mathbf{v}^T \mathbf{V}_S = \frac{\Delta}{2} \begin{pmatrix} \rho+1 & -\sqrt{1-\rho^2} & \\ +\sqrt{1-\rho^2} & \rho-1 & 0 \\ & & \\ 0 & 0 & \end{pmatrix} = \begin{pmatrix} \lambda_S^+ & -\sqrt{\Delta^2-\lambda^2}/2 & \\ \sqrt{\Delta^2-\lambda^2}/2 & \lambda_S^- & 0 \\ & & \\ 0 & 0 & \end{pmatrix}. \tag{49}$$

From Eq (49) we see that the parameter $\Delta$ controls the strength of the coupling between the modes of $\mathbf{J}_S$ through the term $\sqrt{\Delta^2 - \lambda^2}/2$. Thus, in the following analysis, we examine the amplification properties of the system as a function of the parameter $\Delta$.

**Propagator of the dynamics.** We explicitly compute the expression of the propagator for the unit-rank system. From the definition of matrix exponential in terms of infinite sum of matrix powers we obtain

$$\exp(t\Delta\mathbf{u}\mathbf{v}^T) = \sum_{k=0}^\infty \frac{(t\Delta\mathbf{u}\mathbf{v}^T)^k}{k!} = \mathbf{I} + \frac{\Delta\mathbf{u}\mathbf{v}^T}{\lambda}\left(1 + \lambda t + \frac{1}{2}\lambda^2 t^2 + \dots - 1\right) \tag{50}$$

$$= \mathbf{I} + \Delta\frac{e^{\lambda t}-1}{\lambda}\mathbf{u}\mathbf{v}^T.$$

Therefore the final expression for the propagator is given by

$$\mathbf{P}_t = \exp(t(\Delta \mathbf{u}\mathbf{v}^T - \mathbf{I})) = e^{-t} + \Delta e^{-t}\alpha(t,\lambda)\mathbf{u}\mathbf{v}^T, \tag{51}$$

where we introduced

$$\alpha(t,\lambda) = \frac{e^{\lambda t} - 1}{\lambda}. \tag{52}$$

Note that the non-trivial dynamics of the system are restricted to the plane spanned by $\mathbf{u}$ and $\mathbf{v}$. In fact any component of the initial condition orthogonal to this plane decays to zero as $e^{-t}$, as any component orthogonal to $\mathbf{v}$ in the $\mathbf{uv}$-plane. From this it follows that non-monotonic transients occur only if the initial condition of the system has a non-zero component on the structure vector $\mathbf{v}$.

**SVD of the propagator.** To study how the maximum amplification depends on $\Delta$ we compute the amplification envelope $\sigma_1(\mathbf{P}_t)$. The singular values of the propagator $\mathbf{P}_t$ are given by the square roots of the eigenvalues of the matrix $\mathbf{P}_t^T\mathbf{P}_t$. From Eq (51) we can write

$$
\begin{aligned}
e^{2t}\mathbf{P}_t^T\mathbf{P}_t &= (\mathbf{I} + \Delta\alpha(t,\lambda)\mathbf{v}\mathbf{u}^T)(\mathbf{I} + \Delta\alpha(t,\lambda)\mathbf{u}\mathbf{v}^T) \\
&= \mathbf{I} + 2\Delta\alpha(t,\lambda)\mathbf{J}_S + \Delta^2\alpha^2(t,\lambda)\mathbf{v}\mathbf{v}^T.
\end{aligned} \tag{53}
$$

We obtain the expression for the singular values of the propagator $\sigma_{1,2}(\mathbf{P}_t)$ as a function of $\Delta$ and $\lambda$ (see S5 Text):

$$2e^{2t}\sigma_{1,2}^2(\mathbf{P}_t) = 2 + 2\lambda\alpha(t,\lambda) + \alpha^2(t,\lambda)\Delta^2 \pm \sqrt{\Delta^4\left[\alpha^4(t,\lambda) + \frac{1}{\Delta^2}(4\lambda\alpha^3(t,\lambda) + 4\alpha^2(t,\lambda))\right]}. \tag{54}$$

The other $N-2$ singular values of $\mathbf{P}_t$ are equal to $e^{-t}$.

**Choice of the free parameters.** For the unit-rank system, two parameters out of $\Delta$, $\lambda$ and $\rho$ can vary independently. Since we set $\Delta$ as a free parameter, we need to fix the second independent parameter. We explore three scenarios, which imply different scalings of $\lambda$ or $\rho$ with the parameter $\Delta$:

1. keep the eigenvalue $\lambda$ constant, so as to fix the timescale $\tau = 1/(1-\lambda)$, and vary $\Delta$. In this case the correlation $\rho$ between the $\mathbf{u}$ and $\mathbf{v}$ scales according to $\rho = \lambda/\Delta$, meaning that increasing $\Delta$ makes the structure vectors more orthogonal to each other.

2. Fix the correlation between the structure vectors, $\rho$, to a positive value and vary $\Delta$. Increasing $\Delta$ has the effect to increase the timescale of the system $\tau = 1/(1-\Delta\rho)$, until a point where the system becomes unstable, i.e. for $\lambda > 1$, or equivalently $\Delta > 1/\rho$.

3. Keep $\rho$ fixed to a negative value. In this case $\Delta$ can be increased without bounds and higher values of $\Delta$ decrease the timescale $\tau$.

**Maximum amplification of the system.** The singular values of the propagator given by Eq (54) depend in a complex manner on $\Delta$ and $\lambda$. To understand how the maximum amplification of the system depends on $\Delta$, we study the limit of very large $\Delta$, defined as

$$\Delta \gg 2\sqrt{1 - \lambda(\Delta)}, \tag{55}$$

which we call the *strong amplification regime*. Note that in general the eigenvalue $\lambda$ depends on $\Delta$, according to $\lambda(\Delta) = \Delta\rho$. For fixed $\lambda$, Eq (55) is given by $\Delta \gg 2\sqrt{1-\lambda}$, while for a fixed value of $\rho$, Eq (55) translates into $\Delta \gg 2(1-\rho)$ (with the additional constraint $\Delta < 1/\rho$ ensuring

stability, in case $\rho > 0$). If condition given by Eq (55) is met, we can approximate Eq (54) for times $t \gg 2/\Delta$ as

$$2e^{2t}\sigma_1^2(\mathbf{P}_t) \simeq 2 + 2\alpha(t,\lambda)\lambda + 2\alpha^2(t,\lambda)\Delta^2, \qquad t \gg 2/\Delta. \tag{56}$$

For large $\Delta$ we can neglect the first two terms on the right hand side and write the largest singular value as

$$\sigma_1(\mathbf{P}_t) \simeq \Delta e^{-t}\alpha(t,\lambda), \qquad t \gg 2/\Delta. \tag{57}$$

The maximum amplification of the system corresponds to the maximum value in time of $\sigma_1(\mathbf{P}_t)$. In the strong amplification regime (Eq 55) the time $t^*$ at which the singular value attains its maximum is independent of $\Delta$ and reads:

$$t^* = \underset{t}{\mathrm{argmax}} \ e^{-t}\alpha(t;\lambda) = \frac{1}{\lambda}\log\frac{1}{1-\lambda}. \tag{58}$$

Thus, the maximum amplification increases monotonically with $\Delta$:

$$\sigma_1(\mathbf{P}_{t^*}) = g(\lambda(\Delta))\,\Delta, \qquad g(\lambda) = (1-\lambda)^{\frac{1}{\lambda}-1}, \tag{59}$$

where $g(\lambda)$ is a multiplicative factor which depends on the eigenvalue $\lambda$. Different choices of the free parameters imply different growths of the maximum amplification with $\Delta$:

1. for $\lambda$ fixed and $\rho = \lambda/\Delta$, the maximum amplification increases linearly with $\Delta$.

2. For $\rho > 0$ fixed and $\lambda = \Delta\rho$, the maximum amplification increases monotonically with $\Delta$, until it reaches a value equal to $\Delta$ for $\Delta = 1/\rho$ (or $\lambda = 1$).

3. For $\rho < 0$ fixed and $\lambda = \Delta\rho$, the amplification increases monotonically with $\Delta$, but it saturates at a value given by $1/|\rho|$. This follows from the fact that

$$\lim_{\Delta \to +\infty} g(\Delta\rho) = \frac{1}{\Delta|\rho|}. \tag{60}$$

In the case $\rho = 0$ the maximum amplification grows linearly as $\Delta/e$, since

$$\lim_{\rho \to 0} g(\Delta\rho) = \frac{1}{e}. \tag{61}$$

The general observation that the largest eigenvalue of the symmetric part of $\mathbf{J}$ does not provide direct information about the maximum amount of amplification that the system can reach (maximum value of $\|\mathbf{r}(t)\|$ over time) is illustrated by the third case. Here $\rho < 0$ and the largest eigenvalue of the symmetric part of the connectivity is given by $\lambda_{\max}(\mathbf{J}) = \Delta(1 + \rho)/2$. Therefore, while $\lambda_{\max}(\mathbf{J})$ can grow indefinitely by increasing the value of $\Delta$, the maximum amplification saturates at a positive level given by $1/|\rho|$. This indicates that the value of the largest eigenvalue of $\mathbf{J}_S$ is in general inadequate to characterize the maximum amplification of the system, for which other measures may be considered (see Discussion).

**Optimally amplified initial condition and optimal readout.** Using the result we found for the two dimensional case, Eqs (40) and (44), we can determine the angles $\theta_R^* \equiv \theta(\mathbf{R}_1^*)$ and $\theta_L^* \equiv \theta(\mathbf{L}_1^*)$ of the optimal initial condition and optimal readout with respect to the first mode

of $\mathbf{J}_S$ as

$$\tan \theta_{L,R}^* = \pm \arctan \sqrt{\frac{\lambda_S^+ - 1}{1 - \lambda_S^-}} = \pm \sqrt{\frac{\lambda + \Delta - 2}{2 - \lambda + \Delta}}, \tag{62}$$

where the + and − signs correspond respectively to $\theta_L^*$ ans $\theta_R^*$. The optimally amplified initial condition and optimal readout are thus given by

$$\begin{cases} \mathbf{R}_1^* = \cos \theta_R^* \mathbf{x}_S^+ + \sin \theta_R^* \mathbf{x}_S^- \\ \mathbf{L}_1^* = \cos \theta_L^* \mathbf{x}_S^+ + \sin \theta_L^* \mathbf{x}_S^-. \end{cases} \tag{63}$$

Here we examine $\mathbf{R}_1^*$ and $\mathbf{L}_1^*$ in the strong amplification regime (Eq 55). We summarize our results as follows.

1. For fixed $\lambda$ and $\rho = \lambda/\Delta$, we have

$$\tan \theta_R^* \simeq -1 + \frac{2 - \lambda}{\Delta} \tag{64}$$

up to the first order in $\Delta^{-1}$. In the strong amplification regime the second term on the right hand side is much smaller than unity, so that we can compute $\mathbf{R}_1^*$ and $\mathbf{L}_1^*$ at the first order in $\Delta^{-1}$. Denoting by $\mathbf{v}^\perp = (\mathbf{u} - \rho\mathbf{v})/\sqrt{1 - \rho^2}$ and $\mathbf{u}^\perp = (\rho\mathbf{u})/\sqrt{1 - \rho^2}$ respectively the vectors orthogonal to $\mathbf{v}$ and $\mathbf{u}$ in the $\mathbf{uv}$-plane, we can write

$$\begin{cases} \mathbf{R}_1^* \propto \mathbf{v} + \dfrac{1}{2}\dfrac{2 - \lambda}{\Delta}\mathbf{v}^\perp \\ \mathbf{L}_1^* \propto \mathbf{u} + \dfrac{1}{2}\dfrac{2 - \lambda}{\Delta}\mathbf{u}^\perp. \end{cases} \tag{65}$$

In the strong amplification regime the optimal initial condition is thus strongly aligned with $\mathbf{v}$ and the optimal readout with the vector $\mathbf{u}$.

2. For fixed $\rho > 0$ and $\lambda = \Delta\rho$, we compute the value of $\tan\theta^*$ for the largest value $\Delta$ can take before the system becomes unstable, i.e. $\Delta = 1/\rho$. For this value we have

$$\tan \theta_R^* = -\sqrt{\frac{1 - \rho}{1 + \rho}} \simeq -1 + \rho, \qquad \text{for } 0 < \rho \ll 1. \tag{66}$$

Thus we have

$$\begin{cases} \mathbf{R}_1^* \propto \mathbf{v} + \dfrac{\rho}{2}\mathbf{v}^\perp \\ \mathbf{L}_1^* \propto \mathbf{u} + \dfrac{\rho}{2}\mathbf{u}^\perp. \end{cases} \tag{67}$$

3. For fixed $\rho < 0$ and $\lambda = \Delta\rho$, we can write

$$\tan \theta_R^* \simeq -1 + \left(\frac{2}{\Delta} - \rho\right), \tag{68}$$

so that

$$\begin{cases} \mathbf{R}_1^* \propto \mathbf{v} + \dfrac{1}{2}\left(\dfrac{2}{\Delta} - \rho\right)\mathbf{v}^\perp \\[2mm] \mathbf{L}_1^* \propto \mathbf{u} + \dfrac{1}{2}\left(\dfrac{2}{\Delta} - \rho\right)\mathbf{u}^\perp. \end{cases} \tag{69}$$

In conclusion we find that, in the strong amplification regime, the optimal input has a strong component on the structure vector $\mathbf{v}$, while the optimal readout is strongly aligned with $\mathbf{u}$. In cases (2) and (3), however, this requires the additional condition that $\rho$ be small.

### Robustness of the readout to noise in the connectivity

In this section we study the dynamics of the system in presence of noise in the synaptic connectivity. We consider the connectivity matrix given by Eq (41), which implements a single transient pattern, and we add uncorrelated noise of standard deviation $g/\sqrt{N}$ to each weight $\Delta u_i\, v_j$. The resulting connectivity matrix can be written as the sum of a structured unit-rank part and a Gaussian random matrix of the form [35]

$$\mathbf{J} = \Delta \mathbf{u}\mathbf{v}^T + g\boldsymbol{\chi}. \tag{70}$$

The elements of $\boldsymbol{\chi}$ are independently drawn from a Gaussian distribution with zero mean and variance $1/N$ and are uncorrelated with the structured part. In the limit of large $N$, the matrix $\mathbf{J}$ has one eigenvalue equal to the eigenvalue of the unit-rank part, $\lambda = \Delta\rho$, while the other $N-1$ eigenvalues are uniformly distributed in a circle of radius $g$. This holds under the condition that the operator norm of the unit-rank part $\max_{\mathbf{x}}\|\Delta\mathbf{u}\mathbf{v}^T\mathbf{x}\|$ is $O(1)$ [56]. Since the structure vectors $\mathbf{u}$ and $\mathbf{v}$ have unit norm, the operator norm of the unit-rank part is equal to $\Delta$. Therefore, if $\Delta$ is $O(1)$, the condition for the stability of the system is $\max\{\lambda, g\} < 1$.

**Eigenvalues of $\mathbf{J}_S$.** To draw the phase diagram of the system, we compute the eigenvalues of the symmetric part of $\mathbf{J}$

$$\mathbf{J}_S = \Delta\frac{\mathbf{u}\mathbf{v}^T + \mathbf{v}\mathbf{u}^T}{2} + g\boldsymbol{\chi}_S, \qquad \boldsymbol{\chi}_S = \frac{\boldsymbol{\chi} + \boldsymbol{\chi}^T}{2}, \tag{71}$$

where $\boldsymbol{\chi}_S$ denotes the symmetric part of $\boldsymbol{\chi}$. The entries of $\boldsymbol{\chi}_S$ are distributed according to

$$\boldsymbol{\chi}_{S,ij} \sim \begin{cases} \mathcal{N}(0, 1/2N), & i \neq j \\ \mathcal{N}(0, 1/N), & i = j. \end{cases} \tag{72}$$

We can express the eigenvalues of $\mathbf{J}_S$ as a function of $g$ and of the eigenvalues of the symmetric part of the unit-rank matrix (see Eq 44) [57, 58]. In particular, the rightmost eigenvalue of $\mathbf{J}_S$ is given by

$$\lambda_{\max}(\mathbf{J}_S) = \begin{cases} \dfrac{\lambda + \Delta}{2} + \dfrac{g^2}{\lambda + \Delta}, & \lambda + \Delta > \sqrt{2}g \\[2mm] \sqrt{2}g, & \text{otherwise,} \end{cases} \tag{73}$$

where $\sqrt{2}g$ corresponds to the spectral radius of $\boldsymbol{\chi}_S$. We distinguish two cases:

1. if $\sqrt{2}g < 1$, $\lambda_{\max}(\mathbf{J}_S)$ is larger than one only if the two conditions

$$
\begin{cases}
\dfrac{\lambda + \Delta}{2} + \dfrac{g^2}{\lambda + \Delta} > 1 \\
\lambda + \Delta > \sqrt{2}g
\end{cases}
\tag{74}
$$

are satisfied. The first inequality is satisfied if $\lambda + \Delta < 1 - \sqrt{1 - 2g^2}$ or $\lambda + \Delta > 1 + \sqrt{1 - 2g^2}$. Since for $\sqrt{2}g < 1$ we have $1 - \sqrt{1 - 2g^2} < \sqrt{2}g < 1 + \sqrt{1 - 2g^2}$, the condition for the amplified regime becomes

$$
\lambda + \Delta > 1 + \sqrt{1 - 2g^2}.
\tag{75}
$$

2. If $\sqrt{2}g > 1$, the inequality $(\lambda + \Delta)/2 + g^2/(\lambda + \Delta) > 1$ is always satisfied for $\lambda + \Delta > 0$, thus holding also for $\lambda + \Delta > \sqrt{2}g$. From Eq (73) we conclude that, for $\sqrt{2}g > 1$, $\lambda_{\max}(\mathbf{J}_S)$ is larger than one independently of the values of $\lambda$ and $\Delta$.

In the case $\sqrt{2}g < 1$, adding noise in the connectivity has a small effect on the phase diagram of the system. In fact, Eq (75) can be approximated as $\lambda + \Delta \gtrsim 2 - g^2$, which leads to a correction of order $g^2$ to the condition for the amplified regime in absence of noise (see S1 Fig).

**Robustness of the readout activity.** Here we examine the magnitude of the fluctuations around the mean activity introduced by the random term in the connectivity given by Eq (70). In particular we assess the robustness of the readout projection of the response evoked by the optimal stimulus of the noiseless system, i.e. $g = 0$ (for a discussion on the effects of the connectivity noise on the activity orthogonal to the **uv**-plane see S8 Text). For simplicity, we assume that the correlation between the structure vectors, $\rho$, is close to zero, and that the condition for the strong amplification regime is satisfied (Eq (55)). Therefore, the optimal stimulus is strongly aligned with **v**, while the corresponding readout is **u**. We consider the system

$$
\frac{\mathrm{d}r_i}{\mathrm{d}t} = -r_i + \sum_{j=1}^{N} (\Delta u_i v_j + g\chi_{ij})r_j + \sigma\eta_i(t).
\tag{76}
$$

Each neuron receives independent noise with mean zero, variance $\sigma^2$ and autocorrelation function $\langle \eta_i(t)\eta_j(t') \rangle = \delta_{ij}\,\delta(t - t')$, where the angular brackets represent the average over the noise in the input and in the connectivity. In the limit of large $N$, the equation for the mean activity depends only on the structured part of the connectivity:

$$
\frac{\mathrm{d}\langle r_i \rangle}{\mathrm{d}t} = -\langle r_i \rangle + \sum_{j=1}^{N} \Delta u_i v_j \langle r_j \rangle.
\tag{77}
$$

Thus, the mean activity in response to an input along **v** is given by (see Eq 51)

$$
\langle r_i(t) \rangle = e^{-t} v_i + \Delta t e^{-t} u_i
\tag{78}
$$

From Eq (77) we write the equation for the fluctuations of $r_i(t)$ around the mean activity, $\delta r_i(t) = r_i(t) - \langle r_i(t) \rangle$, as

$$
\frac{\mathrm{d}\delta r_i}{\mathrm{d}t} = -\delta r_i + \sum_{j=1}^{N} \Delta u_i v_j \delta r_j + \sum_{j=1}^{N} g\chi_{ij}\langle r_j(t) \rangle + \sigma\eta_i(t),
\tag{79}
$$

where in the third term on the right hand side we neglected the corrections to $r_i(t)$ due to the random component of the connectivity and input noise, keeping only the 0-th order term in $g$, i.e. $\langle r_i(t) \rangle$. Using Eq (78) we can write the solution of Eq (79) as

$$\delta r_i(t) = \sum_{k=1}^{N} \int_0^t \left[ e^{(t-s)(\Delta \mathbf{uv}^T - \mathbf{I})} \right]_{ik} \left[ g \left( \sum_{l=1}^{N} \chi_{kl} \langle r_l(s) \rangle \right) + \sigma \eta_k(s) \right] \mathrm{d}s. \tag{80}$$

The time-dependent correlation matrix $\mathbf{C}(t) = \langle \delta \mathbf{r}(t) \delta \mathbf{r}(t)^T \rangle$ can be written as the sum of two terms, corresponding to the contributions of the noise in the connectivity (with variance proportional to $g^2$) and the noise in the input (with variance $\sigma^2$):

$$\begin{aligned}
C_{ij}(t) \quad &= C_{ij}^g(t) + C_{ij}^\sigma(t) \\
&= \frac{g^2}{N} \sum_{k,l} \int_0^t \int_0^t \mathrm{d}s_1 \mathrm{d}s_2 [e^{(t-s_1)(\Delta \mathbf{uv}^T - \mathbf{I})}]_{ik} [e^{(t-s_2)(\Delta \mathbf{uv}^T - \mathbf{I})}]_{jk} \langle r_l(s_1) \rangle \langle r_l(s_2) \rangle \\
&\quad + \sigma^2 \sum_{k=1}^{N} \int_0^t \mathrm{d}s [e^{(t-s)(\Delta \mathbf{uv}^T - \mathbf{I})}]_{ik} [e^{(t-s)(\Delta \mathbf{uv}^T - \mathbf{I})}]_{jk},
\end{aligned} \tag{81}$$

where in the first term in the right hand side we used $\langle \chi_{kl} \chi_{mn} \rangle = \delta_{km} \delta_{ln} / N$.

We start by computing the first term in Eq (81). Since the elements of the matrix propagator and the mean activity are known (see Eqs 51 and 78), we can compute $C_{ij}^g(t)$ for a given realization of the structured part (see S7 Text). The variance of the activity along the direction of the readout $\mathbf{u}$ due to the noise in the connectivity is computed by projecting the matrix $\mathbf{C}^g$ onto $\mathbf{u}$. In particular we compute the variance of $\delta r_u$ and at the peak of the transient phase ($t^* \simeq 1$, see Eq (58)). As a result, the fluctuations of the readout activity at $t = t^*$ due to the noise in the connectivity read:

$$\mathbf{u}^T \mathbf{C}^g(1) \mathbf{u} = \frac{g^2}{N} e^{-2} \left( \frac{\Delta^4}{36} + \frac{\Delta^2}{2} + 1 \right) \tag{82}$$

and scale as $g\Delta^2 / \sqrt{N}$ (for large $\Delta$).

Computing the variance of the activity along the readout $\mathbf{u}$ due to the input noise yields (see S7 Text)

$$\mathbf{u}^T \mathbf{C}^\sigma(1) \mathbf{u} = \sigma^2 \left[ \frac{1}{2} - \frac{e^{-2}}{2} + \Delta^2 \left( \frac{1}{4} - \frac{5}{4} e^{-2} \right) \right]. \tag{83}$$

From Eq (81), we can write the total amount of variability along the readout $\mathbf{u}$ at the peak amplification as

$$\mathbf{u}^T \mathbf{C}(1) \mathbf{u} = \frac{g^2}{N} e^{-2} \left( \frac{\Delta^4}{36} + \frac{\Delta^2}{2} + 1 \right) + \sigma^2 \left[ \frac{1}{2} - \frac{e^{-2}}{2} + \Delta^2 \left( \frac{1}{4} - \frac{5}{4} e^{-2} \right) \right]. \tag{84}$$

Note that the fluctuations along $\mathbf{u}$ due to the noise in the input do not depend on the size of the network $N$. Therefore, in the limit of large $N$, only the input noise affects the readout activity significantly. By computing the signal-to-noise ratio (SNR) of the readout activity, we can assess the reliability of the readout in presence of input noise. The signal of the readout is simply the amplification level at the peak of the transient phase. Since for orthogonal structure

vectors ($\rho \simeq 0$) the amplification grows as $\Delta/e$, we find

$$SNR(\sigma; \Delta) = \frac{\Delta}{e\,\sigma\sqrt{\left[\frac{1}{2} - \frac{e^{-2}}{2} + \Delta^2\left(\frac{1}{4} - \frac{5}{4}e^{-2}\right)\right]}}. \qquad (85)$$

The readout is reliable if its signal-to-noise ratio is much larger than unity. Interestingly, for large values of $\Delta$ (see Eq 55), the SNR is independent of $\Delta$, so that increasing the amplification does not improve the SNR significantly (see S2 Fig). In fact, for $\Delta \gg 2$, we can approximate Eq (85) as

$$SNR(\sigma; \Delta \gg 2) = \frac{1}{e\,\sigma\sqrt{\frac{1}{4} - \frac{5}{4}e^{-2}}}. \qquad (86)$$

In this regime, the critical value of $\sigma$ above which the $SNR$ becomes smaller than unity is:

$$\sigma_c = \frac{1}{e\sqrt{\frac{1}{4} - \frac{5}{4}e^{-2}}} \simeq 1.17. \qquad (87)$$

We observe that the signal-to-noise ratio along the initial state (the vector $\mathbf{v}$) is simply given by $SNR_0(\sigma) = 1/\sigma$, so that the critical value of $\sigma$ above which $SNR_0$ becomes smaller than unity is given by $\sigma_{c,0} = 1$. As a result, the maximum gain in $SNR$ that strongly amplified networks can achieve is less than 20%. The signal-to-noise ratio along the transient readout $\mathbf{u}$ and along the initial state $\mathbf{v}$ are therefore comparable. However, since $SNR(\sigma;\Delta)<SNR_0$ for non amplified dynamics ($\Delta < 2$), transient amplification is needed to keep a stable $SNR$ across initial state and transient readout.

## Robustness to multiple stored patterns and capacity of the network

In this section we examine the robustness of the transient readouts when $P$ transient trajectories are encoded in the connectivity $\mathbf{J}$. We consider a connectivity matrix given by the sum of $P$ unit-rank matrices

$$\mathbf{J} = \Delta \sum_{p=1}^{P} \mathbf{u}^{(p)}\mathbf{v}^{(p)T}, \qquad (88)$$

where the elements of the vectors $\mathbf{u}^{(p)}$ and $\mathbf{v}^{(p)}$ are randomly and independently distributed with zero mean and variance equal to $1/N$. Therefore, for large $N$ and for $P \leq N/2$, these vectors are close to orthogonal to each other, meaning that the correlation between all the pairs of structure vectors, $\rho$, is close to zero. For simplicity, we assume that the non-normal parameter $\Delta$ is the same for all stored trajectories. We first study the case of two stored transient trajectories ($P = 2$), then generalizing to an extensive number of patterns $P = O(N)$.

**Two encoded transient trajectories.** The connectivity matrix in this case is given by

$$\mathbf{J} = \Delta\mathbf{u}^{(1)}\mathbf{v}^{(1)T} + \Delta\mathbf{u}^{(2)}\mathbf{v}^{(2)T}. \qquad (89)$$

Since the four structure vectors in Eq (89) are uncorrelated with each other, in the limit of large $N$, we can factorize the full propagator of the dynamics as the product of the propagators

of the single unit-rank parts (see Eq (51)) and obtain (see S9 Text)

$$
\begin{aligned}
\exp(t(\mathbf{J} - \mathbf{I})) &\simeq e^{-t}\exp(t\Delta\mathbf{u}^{(1)}\mathbf{v}^{(1)T})\exp(t\Delta\mathbf{u}^{(2)}\mathbf{v}^{(2)T}) \\
&= e^{-t}(\mathbf{I} + \Delta\alpha(t;0)\mathbf{u}^{(1)}\mathbf{v}^{(1)T})(\mathbf{I} + \Delta\alpha(t;0)\mathbf{u}^{(2)}\mathbf{v}^{(2)T}),
\end{aligned}
\tag{90}
$$

where $\alpha(t; \lambda = 0) = t$ (see Eq 52). From Eq (90) we see that, in high dimensionality, the two transient patterns do not interact. In fact, any initial condition defined on the plane spanned by $\mathbf{u}^{(1)}$ and $\mathbf{v}^{(1)}$ evokes a two-dimensional trajectory which remains confined on the same plane. The same holds for the dynamics on the plane defined by $\mathbf{u}^{(2)}$ and $\mathbf{v}^{(2)}$.

**Extensive number of encoded trajectories and capacity of the network.** When the number of encoded trajectories $P$ is of order $N$, we cannot factorize the propagator as in the case of two stored patterns, due to the stronger correlations between the $2P$ structure vectors $\mathbf{u}^{(p)}$ and $\mathbf{v}^{(p)}$. However, the results for the case of one stored pattern with connectivity noise can be applied to this case if we write the connectivity matrix in Eq (88) as

$$
\mathbf{J} = \Delta\mathbf{u}^{(1)}\mathbf{v}^{(1)T} + \Delta\sum_{p=2}^{P}\mathbf{u}^{(p)}\mathbf{v}^{(p)T}.
\tag{91}
$$

Here we isolate the first term of the sum but, since all the $P$ patterns are statistically equivalent, the choice of the first pattern is arbitrary. The vectors $\mathbf{u}^{(i)}$ and $\mathbf{v}^{(i)}$ are uncorrelated with each other, so that we can consider the second term on the right hand side of Eq (91) effectively as noise in the connectivity $\mathbf{J} = \Delta\mathbf{u}^{(1)}\mathbf{v}^{(1)T}$, with mean zero and variance $\Delta^2 P/N^2$. In fact, the mean and the variance of the effective noise are given respectively by

$$
\sum_{p=2}^{P}\langle\mathbf{u}_i^{(p)}\mathbf{v}_j^{(p)}\rangle = \sum_{p=2}^{P}\langle\mathbf{u}_i^{(p)}\rangle\langle\mathbf{v}_j^{(p)}\rangle = 0
\tag{92}
$$

and

$$
\sum_{p,q=2}^{P}\langle\mathbf{u}_i^{(p)}\mathbf{v}_j^{(p)}\mathbf{u}_i^{(q)}\mathbf{v}_j^{(q)}\rangle = \sum_{p,q=2}^{P}\langle\mathbf{u}_i^{(p)}\mathbf{u}_i^{(q)}\rangle\langle\mathbf{v}_j^{(p)}\mathbf{v}_j^{(q)}\rangle = \sum_{p,q=2}^{P}\frac{1}{N^2}\delta_{pq} \simeq \frac{P}{N^2}.
\tag{93}
$$

Applying the results from the previous sections with $g = \Delta\sqrt{P/N}$, we can state that the noise coming from the additional $P-1$ patterns adds fluctuations of the order $\Delta^3\sqrt{P}/N$ to the projection of the activity on the readout $\mathbf{u}^{(1)}$ corresponding to the stimulus $\mathbf{v}^{(1)}$. Since the number of encoded patterns $P$ is extensive, the readout fluctuations scale as $1/\sqrt{N}$.

However, when a number $P$ of trajectories are encoded in $\mathbf{J}$, we are not guaranteed that the connectivity has stable eigenvalues. Indeed, the eigenvalues of the matrix $\Delta\sum_{p=2}^{P}\mathbf{u}^{(p)}\mathbf{v}^{(p)T}$ are distributed in a circle of radius $g = \Delta\sqrt{P/N}$ (yet the spectral density is not uniform, since Eq (88) can be written as the product of two rectangular Gaussian matrices) [59]. Thus, to ensure overall stability we need $g = \Delta\sqrt{P/N} < 1$, resulting in a maximal number of patterns $P_{\text{max}}$ that can be stored in the connectivity before the system becomes unstable. This number defines the capacity of the system and is given by

$$
P_{\text{max}} = \frac{1}{\Delta^2}N.
\tag{94}
$$

If the vectors $\mathbf{u}^{(p)}$ and $\mathbf{v}^{(p)}$ are exactly orthogonal to each other for all $p$, therefore forming an orthonormal basis in $\mathbb{R}^N$, Eq (94) reduces to $P_{\text{max}} = N/2$. From Eq (94) we see that,

for fixed Δ, the number of transient trajectories that we can encode in the connectivity matrix scales linearly with the size of the system, $N$. The capacity of the system rapidly drops when Δ is increased, meaning that more amplified systems can encode less number of stimuli. When the structure vectors are orthogonal to each other as in our case ($\rho \simeq 0$), the system is amplified for $\Delta > 2$ (see Eq 45). Therefore, Eq (94) evaluated at $\Delta = 2$ provides an upper bound on the capacity for an amplified system with uncorrelated structure vectors:

$$P_{\max} < 0.25\, N. \tag{95}$$

## Supporting information

**S1 Text.**
(PDF)

**S2 Text.**
(PDF)

**S3 Text.**
(PDF)

**S4 Text.**
(PDF)

**S5 Text.**
(PDF)

**S6 Text.**
(PDF)

**S7 Text.**
(PDF)

**S8 Text.**
(PDF)

**S9 Text.**
(PDF)

**S1 Fig. Phase diagram for the unit-rank network with connectivity noise. A**. $g < 1/\sqrt{2}$. The red line indicates the boundary between the monotonic and amplified parameter regions for $g = 0.5$. The grey dashed line corresponds to the case $g = 0$. **B**. $g > 1/\sqrt{2}$. The dynamics are amplified regardless of the values of the parameters Δ and $\rho$.
(TIF)

**S2 Fig. Signal-to-noise ratio of the readout in presence of external input noise.** Signal-to-noise ratio of the readout as a function of the standard deviation of the input noise $\sigma$ for two values of the non-normal parameter Δ. Non-amplified dynamics ($\Delta = 1$) are less robust to noise than amplified dynamics ($\Delta = 4$). Dashed lines correspond to the theoretical values (Eq 85). In simulations, $N = 1000$. Errorbars represent the standard deviation of the mean over 200 realizations of the connectivity matrix.
(TIF)

## Acknowledgments

We are grateful to Francesca Mastrogiuseppe and Manuel Beiran for discussions and feedback on the manuscript.

## Author Contributions

**Conceptualization:** Giulio Bondanelli, Srdjan Ostojic.

**Data curation:** Giulio Bondanelli, Srdjan Ostojic.

**Formal analysis:** Giulio Bondanelli, Srdjan Ostojic.

**Funding acquisition:** Srdjan Ostojic.

**Investigation:** Giulio Bondanelli, Srdjan Ostojic.

**Methodology:** Giulio Bondanelli, Srdjan Ostojic.

**Project administration:** Srdjan Ostojic.

**Resources:** Giulio Bondanelli, Srdjan Ostojic.

**Software:** Giulio Bondanelli.

**Supervision:** Srdjan Ostojic.

**Validation:** Giulio Bondanelli, Srdjan Ostojic.

**Visualization:** Giulio Bondanelli, Srdjan Ostojic.

**Writing – original draft:** Giulio Bondanelli, Srdjan Ostojic.

**Writing – review & editing:** Giulio Bondanelli.

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
