## [Decision Letter · Decision Letter 0]

14 Nov 2019

Dear Dr Bondanelli,

Thank you very much for submitting your manuscript, 'Coding with transient trajectories in recurrent neural networks', to PLOS Computational Biology. As with all papers submitted to the journal, yours was fully evaluated by the PLOS Computational Biology editorial team, and in this case, by independent peer reviewers. The reviewers appreciated the attention to an important topic but identified some aspects of the manuscript that should be improved.

We would therefore like to ask you to modify the manuscript according to the review recommendations before we can consider your manuscript for acceptance. Your revisions should address the specific points made by each reviewer and we encourage you to respond to particular issues Please note while forming your response, if your article is accepted, you may have the opportunity to make the peer review history publicly available. The record will include editor decision letters (with reviews) and your responses to reviewer comments. If eligible, we will contact you to opt in or out.raised.

- Supporting Information uploaded as separate files, titled 'Dataset', 'Figure', 'Table', 'Text', 'Protocol', 'Audio', or 'Video'.

We hope to receive your revised manuscript within the next 30 days. If you anticipate any delay in its return, we ask that you let us know the expected resubmission date by email at ploscompbiol@plos.org.

Sincerely,

Kenneth D. Miller

Guest Editor

PLOS Computational Biology

Lyle Graham

Deputy Editor

PLOS Computational Biology

[LINK]

The reviewers agree that this paper is a good and solid contribution but also raise many smaller issues that need to be addressed. Their reviews are clear, I would add just two very tiny comments to reviewer comments: reviewer 2 comment 1: if there are degenerate eigenvalues, then either the corresponding subspace must be missing an eigenvector, or that subspace of eigenvectors is normal (an orthonormal basis of eigenvectors can be chosen for that subspace). reviewer 3 comment 3: more generally for a normal matrix the singular values are the absolute values (or modulus, for complex eigenvalues) of the eigenvalue- singular values are nonnegative and real, eigenvalues can be negative or complex.

Reviewer's Responses to Questions

**Comments to the Authors:**

Reviewer #1: This paper provides a number of useful analytical results characterizing the strength and structure

of non-normal transient amplification, which is of relevance to the study of dynamics in many biological networks,

including recurrent neural networks.

In the theoretical neuroscience literature, transient amplification has been previously proposed as a mechanism

underlying fast spontaneous cortical fluctuations and selective amplification of neural patterns, as well as in

networks with extensive working memory.

While non-normality (non-orthogonality of eigenvectors) is necessary for transient amplification in linear networks,

it is clear that any slight deviation from normality is not sufficient for creating amplification.

The authors therefore present sufficient conditions for the latter (criteria in terms of eigenvalues of the symmetrized

connectivity matrix, which have been known previously, at least in the math literature on transient amplification),

which they also specialize to classes of network structures that have been of interest in theoretical neuroscience.

They then provide a simple method for determining the spatial (i.e. in terms of neural population pattern) and temporal structure of transient amplification using a singular value decomposition of the linear network’s propagator.

They use these tools to study transient amplification in effectively low dimensional (i.e. low-rank) as well as high dimensional random systems, and to study the robustness of selective amplification in the presence of random structural and dynamic perturbations. Finally they provide a proposal for constructing systems that can selectively (and transiently) amplify multiple specific patterns, and they quantify the corresponding capacity of such linear networks —i.e. how many on-average orthogonal patterns can be “learned” (i.e. included in the connectivity matrix), without disrupting the desired selective amplification.

This is a nice paper with several technical results that could be quite useful to the theoretical/computational

neuroscience community. Even though many of these results are easy to derive —and a few have been already published, as noted by authors, in the math literature on transient amplification— I still believe this is a useful paper for the community, in that it presents all such old and new results together, specializes them to some network structures of interest in neuroscience, and also makes proposals about their possible computational utility.

The paper is also well-written, and discussions and expositions were clear and to-the-point. So I recommend its publication with minor corrections that I list below.

Comments:

1. Lines 93-94 (and also lines 162): The language needs to be refined/made more precise. It’s, strictly speaking, meaningless to talk about identifying “different inputs giving rise to amplified trajectories” or estimating “their numbers”, or asking “… how many inputs are amplified?", since the input space (and its subspace with those properties) is a continuum, or more precisely a vector space with infinite members. You should rather say “identity the input subspace giving rise to amplified trajectories” and “estimating the dimensionality of this subspace”, and/or “how many orthogonal inputs [or input patterns] are amplified?".

2. Lines 126-127: you say “… except in the case of 126 one-dimensional dynamics or symmetric connectivity matrices.” The exception is not limited to symmetric matrices and should cover all normal J’s, so this phrase should be corrected. (E.g. consider anti-symmetric J’s which have imaginary eigenvalues: in that case the symmetric part is 0 and all its eigenvalues are 0, so the second condition is 0 >1, while because eigenvalues of J are imaginary the first condition is 0 < 1, and the two are inconsistent.)

3. Line 210: you end the sentence by “… when the connectivity J is Gaussian.", but the correct statement is “… when the connectivity matrix J is a random matrix with independent and identically distributed elements.” Gaussianity is not key (due to universality theorems on the circle and semicircular laws), but independence is key: so conversely a (symmetric) gaussian matrix with nontrivial covariances across elements need not give rise to the circle (semicircular) law.

4. Caption of Fig. 4: Change the caption for panel C to "C. Illustration of the dynamics elicited by the three inputs, R_1, R_2 and R_100 (show in different rows), as in B.”

5. Lines 228 and 231: you need to define what you mean by minimal connectivity. It seems that you mean a matrix with minimal rank. So try to justify why that notion of minimality is relevant to neuroscience.

6. Lines 280-292: the writing has to be made accurate regarding the “orthogonality" of different pattens. If the u_p/v_p vectors pairs for different p’s are literally mutually orthogonal (as the current text suggests) then there will be exactly zero interference between different rank-one terms (i.e. different patterns), and fluctuations of the activity along a readout u_p caused by other patterns will be exactly 0, and not just O(\\sqrt{P}/N). In this case the “capacity” would be N/2. But reading the Methods section it becomes clear that the authors are implicitly talking about a connectivity with the rank-P form with the different u’s and v’s sampled from a random ensemble of vectors with independently distributed components, such that different vectors are only approximately orthogonal with high probability (when N is large), and not literally orthogonal as the main text in lines 280-290 suggests. So this should be corrected.

7. Lines 27-28: replace “… but sufficient conditions for such amplification were not given.” with “… but general sufficient conditions for such amplification were not given.”

8. Line 53: I would add the following at the end of this sentence (first sentence of “Monotonic vs Amplified Transient Trajectories”): “… in the autonomous network with I(t) = 0."

9. Lines 740, 749 and 763: fix the broken (appearing as ?) references.

10. The brackets (signifying averaging) around dr_i on the left hand side should be removed.

11. In equation 80 (lines 782-783): the sum over the index l is only needed for the first term and the second term (the one involving \\eta_k) should not be summed over l.

12. Line 817: “randomly distributed” should be replaced with “randomly and independently distributed”.

13. Line 343: “straightforward” doesn’t have a hyphen.

14. Figure 6: In panel E, the maximum readout value for P/N ~ 0.02 seems to be significantly higher than the maximum of the red curve in panel D. Why?

15. Figure 2: in panel B, it would be helpful to also include a line for \\tau>1, corresponding to max Re \\lambda(J) >0.

16. Figure 2: panel C would be more inofrmative if it is turned into a heat-map of \\sigma_1^*, with the two phases (monotonic and amplified) and their boundary indicated/drawn on top of the heat-map.

17. In caption of Fig. 4, panel B: replace “… at time t_* in pannel [sic.] A” with “… at time t_* indicated by the dashed vertical line in panel A”.

18. Fig 4 C, last row, middle column: unlike in the top and middle row of panel C, the labels L_100^* and R_100^* are missing here.

19. Line 544: replace "symmetrically arranged along the imaginary dimension.” with "symmetrically arranged on either side of the real axis.”

,

Reviewer #2: The authors provide a clear and elegant explanation of the behavior of linear networks of neurons in terms of transient amplification. The paper establishes the basic tools needed to understand transient amplification (eigenvalues of the symmetric part of the connectivity matrix, SVD of the propagator…) and analyses in details random and low-rank connectivity matrix, together allowing the reader to develop an intuitive understanding of the phenomenon. I also liked very much the structuring of the paper into 3 parts: simple take-home messages, methods, and appendices. I congratulate the authors. The study will be very useful for teaching.

I only have a couple of small comments:

1) I had the intuition that strong amplification required not only non-normality (lack of orthogonality of the eigenvectors of J), but also a **difference** between their corresponding eigenvalues. In this case, initial conditions loading on several non-orthogonal eigenvectors (i.e., those involving ‘cancellations’) align initially with the slowest decaying modes (this is the period of growth), and eventually decay. Is this intuition incorrect in general, or is this picture somehow hidden in the properties of J_s?

2) Transient amplification (TA) is usually studied in connection to specific initial conditions from which amplification happens. The ‘transient’ is then purely generated by the recurrent connectivity. From the point of view of neuroscience, I think it would be interesting to include some comments in the discussion about the kinds of experimental or real-life situations where this may be expected to occur or not. For instance, TA has been related to preparatory states before movement. How about onset sensory responses in cortex? How about time-varying inputs (which actually are the ethological norm)? The discussion on this would benefit from the recognition that, although linear models provide interesting approximations, the real networks have significant non-linearities, even at the firing rate level… (I say this because if everything is purely linear, there is perfect superposition in time and time-varying input is not particularly interesting...)

Reviewer #3: # Summary and main comments

This paper discusses non-normal mechanisms for transient amplification in linear recurrent networks. The authors study the distinction between two classes of networks: those that transiently amplify their inputs, and those that do not. They provide a necessary and sufficient condition on the connectivity matrix for a network to belong to the latter, and go on explaining i) how to find those inputs that are transiently amplified and ii) how to construct networks that can exploit such transient amplification effects to perform "coding".

I am very strongly supportive of publication, as this is both a timely and very well executed paper, easy to read, well illustrated. I only have a couple of suggestions:

1. Perhaps my main comment concerns the old debate on matrix nonnormality, and how it is inherently difficult to characterise a concept that has to do with the geometry of a matrix' eigenbasis using a single scalar number (here, the numerical abscissa). In this paper, the distinction is made between networks in which the activity vector can grow transiently in response to an appropriately chosen input; and those that simply can't, regardless of the input. The authors derive a diagnostic criterion based on the numerical abscissa; while this is entirely valid (given the definition of "amplifying vs non-amplifying" networks outlined above), it is nevertheless an asymptotic criterion which only concerns the slope of the norm $\\| r(t)\\|$ of the response at small times ($t=0^+$) and therefore _in principle_ provides little insight about how much $\\| r(t) \\|$ will grow and for how long. To be fair, the authors go on discussing an analysis of the induced 2-norm of the propagator $e^{tA}$, which gives the full description of transients, but ─ as they acknowledge ─ whose SVD can only be computed numerically in most cases. There is a whole chapter in Trefethen & Embree ("Spectra and pseudospectra"; chap. 4 on transients) which discusses alternative summary scalar measures that are much better indications of expected transient magnitude / durations (i.e. provide useful bounds on transient size & timing). The pseudospectrum and associated Kreiss theorems should probably be mentioned.

So I suggest the authors discuss these issues briefly in the intro / discussion. Perhaps a useful place to start is the 2x2 example they have already studied in depth, for which it's not too difficult to find corner cases where the max amplification is bounded whereas the numerical abscissa isn't. E.g. the peak amplification for W = [ 1; -a]; [1; -a] ] is upper bounded by $\\sqrt{2}$ (it tends to $\\sqrt{2}$ as $a \\to \\infty$); yet the numerical abscissa (spectral abscissa of $W+W^T-2I$) grows linearly in $a$ ($\\approx (\\sqrt(2)-1) a$ (the peak of amplification occurs earlier and earlier, too).

2. Going back to the "coding" viewpoint, adopted in the introduction: for the type of deterministic Markovian dynamics considered here, future states are entirely determined by the current one ─ the whole trajectory is entirely determined by the initial condition; so for deterministic dynamics, there can be no advantage of eliciting fancy transients: there is just as much information about the stimulus at the peak of the transient as there was at $t=0$. The advantage of amplification for coding become apparent when noise is taken into consideration ─ but then, it is important to distinguish between process noise (which tends to get amplified together with signal) and observation noise (which doesn't). Nonnormal transients don't help much with process noise (though that depends on its geometry), but can hugely increase the mutual information between stimulus and r(t) in cases when output (observation) noise is strong; having signal rise above noise is a big win. I would recommend discussing this briefly, too.

# Minor comments:

1. line 30:

> This results leads to a simple distinction between two classes of networks: networks in which all inputs lead to weak, decaying transients, and networks in which specific inputs elicit strongly amplified transient responses.

Cf main comment (1) above: this is a little too strong, as the class of networks of the latter type also contains networks that amplify very weakly.

2. line 71:

> necessarily implies that the firing rate of at least one neuron shows a transient increase before decaying to baseline.

increase, or decrease (but can always be turned into an increase by reversing the sign of the initial condition of course)

3. line 177:

> Note that for a normal matrix, the left and right singular vectors $R(t)_k$ and $L(t)_k$ are identical, and the singular values are equal to the eigenvalues [...]

corner case: e.g. skew symmetric (hence normal) matrix W has complex eigenvalues, whereas singular values are real

4. regarding the margin:

> To eliminate the trajectories with very short amplification, one can further constrain the slopes to be larger than a margin $\\epsilon$

again, this asymptotic result concerns the slope at time t=0; even with a margin, this seems to give no formal guarantee on the amount of amplification that can follow

5. line 228:

> Specifically, we wish to determine the minimal connectivity that transiently transforms a fixed, arbitrary input $r_0$ into a fixed, arbitrary output $w$, through two-dimensional dynamics.

Considerations of time unclear here → transform $r_0$ into $w$ _at some point_ during the transient?

6. reference missing on line 405

7. reference to figure missing on line 808.

Best wishes,

Guillaume Hennequin

**Have all data underlying the figures and results presented in the manuscript been provided?**

Reviewer #1: Yes

Reviewer #2: Yes

Reviewer #3: Yes

PLOS authors have the option to publish the peer review history of their article (what does this mean?). If published, this will include your full peer review and any attached files.

Reviewer #1: No

Reviewer #2: No

Reviewer #3: Yes: Guillaume Hennequin

---

## [Editor Report · Decision Letter 1]

3 Jan 2020

Dear Dr Bondanelli,

Thank you very much for submitting your manuscript, 'Coding with transient trajectories in recurrent neural networks', to PLOS Computational Biology. As with all papers submitted to the journal, yours was fully evaluated by the PLOS Computational Biology editorial team, and in this case, by independent peer reviewers. The reviewers appreciated the attention to an important topic but identified some aspects of the manuscript that should be improved.

We would therefore like to ask you to modify the manuscript according to the review recommendations before we can consider your manuscript for acceptance. Your revisions should address the specific points made by each reviewer and we encourage you to respond to particular issues Please note while forming your response, if your article is accepted, you may have the opportunity to make the peer review history publicly available. The record will include editor decision letters (with reviews) and your responses to reviewer comments. If eligible, we will contact you to opt in or out.raised.

- Supporting Information uploaded as separate files, titled 'Dataset', 'Figure', 'Table', 'Text', 'Protocol', 'Audio', or 'Video'.

We hope to receive your revised manuscript within the next 30 days. If you anticipate any delay in its return, we ask that you let us know the expected resubmission date by email at ploscompbiol@plos.org.

Sincerely,

Kenneth D. Miller

Guest Editor

PLOS Computational Biology

Lyle Graham

Deputy Editor

PLOS Computational Biology

[LINK]

(From K Miller, guest editor)

The reviews supported publication of the paper and the authors have responded well to all the review comments. The paper is almost ready for publication. However in scanning through the paper looking at the corrections I noticed a number of very small points that should be revised in a final version:

Abstract: "and networks in which specific inputs elicit strongly amplified transient responses and are mapped onto orthogonal output states during the dynamics". Here you are talking about the whole class of networks that show initial amplification, which may not be strongly amplified and need not be mapped onto orthogonal states, so this should be changed, e.g. to "and networks in which specific inputs elicit amplified responses during the dynamics". You could if you want restore the orthogonal concept by inserting 'orthogonal' in "We then build minimal, low-rank networks ... mapping a specific input onto a specific *orthogonal* output state."

Line 62: monotonic decay "exploring essentially a single dimension" -- this isn't true, the monotonic decay could curve or spiral through an arbitrarily high-dimensional space. So I suggest cutting this phrase. 'or transiently move away from it by following a rotation': similarly, this need not be a rotation -- for example if in 2D you have two eigenvectors both nearly vertical, and you start with a small horizontal initial condition, and eigenvalues are real and of very different magnitudes, then the trajectory will go up toward the eigenvector with the larger eigenvalue and then decay down along that eigenvector, which is nearly an up and down motion with only a slight rotational component. So I'd suggest cutting "by following a rotation".

Line 72: "the firing rate of at least one neuron shows a transient increase (or decrease) before decaying to baseline" -- 'increase or decrease' includes pretty much everything, so this is more or less a tautology. I think what you want to state is that at least one r_i shows a transient increase in its absolute value.

You've also gotten yourself into a bit of trouble since you are calling the r_i's the deviation of the firing rate from the fixed point, so you cannot simply refer to them as 'the firing rates', i.e. you can't say 'at least one firing rate increases in its absolute value'. You could get yourself out of this by, under Eq. 1, saying that for simplicity you will refer to the r_i's as the firing rates, in which case you would want to say 'before decaying to zero' instead of 'before decaying to the baseline'.

Lines 144-147: "two interacting excitatory-inhibitory populations" is confusing, it sounds like 4 populations, two excitatory and two inhibitory. You could just say 'an excitatory and an inhibitory population'.

"our criterion states that the excitatory feedback needs to be (approximately) larger than unity in order to achieve transient amplification (Fig. 2 and Methods)". 'Methods' should instead be Appendix F. But also, this statement isn't correct -- it's only correct when the network is of the form {{w,-k w},{w,-k w}} with k=1+\\epsilon for \\epsilon small. It is certainly not true for the general case of one excitatory and one inhibitory population. The same problem is in the legend of Figure 2, which again states this same criterion, now the form of the matrix is specified but you haven't noted the restriction on k.

Lines 315 and 323: I think you want to cite citation [37], not [38]

Lines 378-380: "the inhibition-dominated regime, which as we show approximately corresponds to the class of unit-rank E-I networks satisfying the general criterion for transient amplification." This isn't true -- if you assume the form {{w,-k w},{w,-k w}}, you can get transient amplification in a stable network either with k<1 or k>1.

(Mathematica tells me: If k<1, the criterion is (2 (-1 + k))/(1 + k)^2 + 2 Sqrt[2] Sqrt[(1 + k^2)/(1 + k)^4] < w < -(1/(-1 + k)); while if k>1, the criterion is w > (2 (-1 + k))/(1 + k)^2 + 2 Sqrt[2] Sqrt[(1 + k^2)/(1 + k)^4].)

Methods, Eq. 30: You should note that, *for J stable*, Eq 30 is the criterion for Eq. 28 to be >1, and that the stability condition that Det(J-1)>0 is precisely the condition that the argument of the square root in (30) is positive, so that \\Delta_c is real. Without the condition that J is stable, the criterion for Eq 28 > 1 is more complicated.

Appendix F, line 1012 "recovers the results from (32), showing that the system is amplified if the excitatory strength w is (approximately) larger than one." In (32), we showed something different: that for an initial condition of r_E>0, r_I=0, r_E (not |r|) shows transient amplification if and only if w>1, for any value of k. What you are showing is that there will be some initial condition for which |r| will show transient amplification if k=1+\\epsilon and w>1-\\epsilon^2/4 for \\epsilon<<1. These are very different.

I'm sorry it took me a while to get to this after the revision came in, due to the holidays. If it comes back with these revisions I promise very quick turnaround.

---

## [Editor Report · Decision Letter 2]

14 Jan 2020

Dear Dr Bondanelli,

We are pleased to inform you that your manuscript 'Coding with transient trajectories in recurrent neural networks' has been provisionally accepted for publication in PLOS Computational Biology.

In the meantime, please log into Editorial Manager at https://www.editorialmanager.com/pcompbiol/, click the "Update My Information" link at the top of the page, and update your user information to ensure an efficient production and billing process.

One of the goals of PLOS is to make science accessible to educators and the public. PLOS staff issue occasional press releases and make early versions of PLOS Computational Biology articles available to science writers and journalists. PLOS staff also collaborate with Communication and Public Information Offices and would be happy to work with the relevant people at your institution or funding agency. If your institution or funding agency is interested in promoting your findings, please ask them to coordinate their releases with PLOS (contact ploscompbiol@plos.org).

Thank you again for supporting Open Access publishing. We look forward to publishing your paper in PLOS Computational Biology.

Sincerely,

Kenneth D. Miller

Guest Editor

PLOS Computational Biology

Lyle Graham

Deputy Editor

PLOS Computational Biology

---

## [Editor Report · Acceptance letter]

3 Feb 2020

PCOMPBIOL-D-19-01144R2 

Coding with transient trajectories in recurrent neural networks

Dear Dr Bondanelli,

I am pleased to inform you that your manuscript has been formally accepted for publication in PLOS Computational Biology. Your manuscript is now with our production department and you will be notified of the publication date in due course.

With kind regards,

Laura Mallard
